# Markers of Angiogenesis, Lymphangiogenesis, and Epithelial–Mesenchymal Transition (Plasticity) in CIN and Early Invasive Carcinoma of the Cervix: Exploring Putative Molecular Mechanisms Involved in Early Tumor Invasion

**DOI:** 10.3390/ijms21186515

**Published:** 2020-09-06

**Authors:** Olga Kurmyshkina, Pavel Kovchur, Ludmila Schegoleva, Tatyana Volkova

**Affiliations:** 1Institute of High-Tech Biomedicine, Petrozavodsk State University, 185910 Petrozavodsk, Russia; olya.kurmyshkina@yandex.ru; 2Department of Hospital Surgery, ENT Diseases, Ophthalmology, Dentistry, Oncology, Urology, Institute of Medicine, Petrozavodsk State University, 185910 Petrozavodsk, Russia; pkovchur@mail.ru; 3Department of Applied Mathematics and Cybernetics, Institute of Mathematics and Information Technologies, Petrozavodsk State University, 185910 Petrozavodsk, Russia; schegoleva@petrsu.ru

**Keywords:** angiogenesis, lymphangiogenesis, epithelial-mesenchymal transition, tumor invasion, cervical intraepithelial neoplasia, cervical cancer, molecular markers, signaling pathways

## Abstract

The establishment of a proangiogenic phenotype and epithelial-to-mesenchymal transition (EMT) are considered as critical events that promote the induction of invasive growth in epithelial tumors, and stimulation of lymphangiogenesis is believed to confer the capacity for early dissemination to cancer cells. Recent research has revealed substantial interdependence between these processes at the molecular level as they rely on common signaling networks. Of great interest are the molecular mechanisms of (lymph-)angiogenesis and EMT associated with the earliest stages of transition from intraepithelial development to invasive growth, as they could provide the source of potentially valuable tools for targeting tumor metastasis. However, in the case of early-stage cervical cancer, the players of (lymph-)angiogenesis and EMT processes still remain substantially uncharacterized. In this study, we used RNA sequencing to compare transcriptomes of HPV(+) preinvasive neoplastic lesions and early-stage invasive carcinoma of the cervix and to identify (lymph-)angiogenesis- and EMT-related genes and pathways that may underlie early acquisition of invasive phenotype and metastatic properties by cervical cancer cells. Second, we applied flow cytometric analysis to evaluate the expression of three key lymphangiogenesis/EMT markers (VEGFR3, MET, and SLUG) in epithelial cells derived from enzymatically treated tissue specimens. Overall, among 201 differentially expressed genes, a considerable number of (lymph-)angiogenesis and EMT regulatory factors were identified, including genes encoding cytokines, growth factor receptors, transcription factors, and adhesion molecules. Pathway analysis confirmed enrichment for angiogenesis, epithelial differentiation, and cell guidance pathways at transition from intraepithelial neoplasia to invasive carcinoma and suggested immune-regulatory/inflammatory pathways to be implicated in initiation of invasive growth of cervical cancer. Flow cytometry showed cell phenotype-specific expression pattern for VEGFR3, MET, and SLUG and revealed correlation with the amount of tumor-infiltrating lymphocytes at the early stages of cervical cancer progression. Taken together, these results extend our understanding of driving forces of angiogenesis and metastasis in HPV-associated cervical cancer and may be useful for developing new treatments.

## 1. Introduction

The establishment of a proangiogenic phenotype and epithelial-to-mesenchymal transition (EMT) are presently considered as critical events that promote the induction of invasive growth in epithelial tumors, and stimulation of lymphangiogenesis is believed to confer the capacity for early metastatic dissemination to cancer cells [1]. Recent research has provided convincing evidence that, at the molecular level, the mechanisms of angiogenesis, lymphangiogenesis, and EMT have the same trigger factors, and that the signaling pathways regulating them overlap significantly; therefore, they can be (or even should be) explored as a single process when profiling gene expression in primary tumors. The ability of many molecular players that control the mechanisms of tumor-associated (lymph-)angiogenesis and EMT to compensate for each other’s functions enables their extremely high plasticity. This can apparently explain the fact that anti-angiogenic (anti-VEGF/VEGFR) therapy, despite its suppressive effect on primary tumor growth, may result in promotion of metastatic spread via compensatory upregulation of genes mediating EMT and lymphangiogenesis. Conversely, simultaneous blockade of the lymphangiogenic/EMT regulators inhibits migration capacity of cancer cells [2,3], highlighting the importance of comprehensive consideration of triggering mechanisms of invasion and metastasis. In this regard, of great interest are the (lymph-)angiogenesis and EMT-governing molecular factors and pathways associated with transition of intraepithelial lesion to invasive behavior [4], the earliest stages defined, according to morphological criteria, as carcinoma in situ (CIS) and carcinoma “with minimal stromal invasion”, or microinvasive cancer [5]. Although opportunities for cancer detection at these early stages of progression are still limited, there are a growing number of studies investigating molecular alterations across the entire human genome (including mutation landscape analysis, comparative transcriptome profiling, and genome-wide screening of epigenetic aberrancies) in precancerous lesions and early-stage invasive cancer with the use of high-throughput techniques. Importantly, the potential of Next-Generation Sequencing-based techniques such as RNA sequencing (RNA-Seq) applied to different transitional states in the early phases of cancer development lies in the opportunity to get a better and more comprehensive insight into advanced metastatic cancer [4].

With respect to virus-associated cancers, which include human papillomavirus (HPV)-related cancers, few data are available yet. For example, Masterson and co-authors employed RNA-Seq to compare whole transcriptome profiles of HPV-positive and HPV-negative early-stage oropharyngeal carcinoma samples from a prospective cohort of patients followed by qPCR validation of the results using RNA extracted from adjacent areas of carcinoma in situ and invasive carcinoma microdissected from formalin-fixed paraffin-embedded (FFPE) sections [6]. Regarding cervical cancer, which development is predominantly caused by HPV infection, its preclinical forms, which in turn include cervical intraepithelial neoplasia (CIN of grades 1–3, CIN1–3), cancer in situ (CIS, stage “0”) and microinvasive cancer (stage IA, according to the International Federation of Gynecology and Obstetrics, FIGO, staging system), make up a significant, as compared to other types of cancer, proportion of cases (Figure 1). However, at the same time cervical cancer cells exhibit extremely aggressive properties and have high metastatic potential. This fact allows one to consider cervical cancer to be an important object for genome-wide investigations of molecular factors that drive (lymph-)angiogenesis and EMT to sustain early steps of invasive growth. The importance of this issue has been emphasized in the study by Jung et al. [7] who performed whole-exome sequencing of CIN3/CIS, microcarcinoma, and frank invasive cervical cancer of IB−IIB stages to describe changes in mutation profile which occur as intraepithelial lesions progress to true invasive cancer. From the assessment of “evolutionary ages”, the authors infer that microinvasive carcinoma does represent an intermediate stage between CIN/CIS and invasive cancer. However, the types and the frequencies of mutations (or copy number alterations) found in driver genes and other gene loci suggest that microcarcinoma corresponds to invasive cancer, being thereby dramatically different from CIN3/CIS, with the recurrent mutations identified shown to affect cancer-related signalings that control cell adhesion and migration. In another study, RNA-Seq transcriptome analysis of co-existing low- and high-grade squamous intraepithelial lesions performed on FFPE samples made possible characterization of a set of differentially expressed genes that might contribute to the promotion of CIN progression from low-grade to high-grade lesions [8]. Two more recently published studies by Lin et al. [9,10] are devoted to RNA-Seq-based analysis of mRNA and microRNA transcriptomes and the related signaling pathways in invasive cervical cancer of FIGO IB, IIA, and IIB stages, which are sometimes referred to as early-stage and locally advanced cancer. Although these studies provide comprehensive descriptions of gene activity profiles supporting the progressive nature of molecular dysregulations, one can see there is little other published NGS-based information on putative factors that launch (lymph-)angiogenesis and EMT during the transition from intraepithelial development to invasive growth. At the same time, the functionality of both types of vessels (blood and lymphatic) and synergistic impact of angiogenesis and lymphangiogenesis on cervical cancer metastasis, as well as the specific pattern of gene activity has been experimentally confirmed using different approaches, for example, patient derived xenograft models of cervical cancer [11] or co-cultures of cervical cancer cells and human umbilical vein endothelial cells [12], thus providing a rationale for developing new strategies for targeted anti-metastatic therapies.

Building on their cDNA microarray data, Gius et al. previously proposed distinctive gene expression signatures for different steps of CIN progression to an invasive state, with the “angiogenic signature” documented at the stage of <CIN3 [13]. Taking into account RNA-Seq capabilities, in the present work we considered it necessary to continue exploration of biomarkers of angiogenesis, lymphangiogenesis, and EMT, and the relationships between them in the context of intraepithelial and early-stage invasive carcinoma of the cervix (Figure 1); as, being associated with cells gaining an invasive migratory phenotype, these molecular factors are thought to play pivotal roles in the onset of cervical cancer metastasis, either via lymphatic or hematogenous routes.

## 2. Results

### 2.1. Differential Gene Expression Analysis

To compare the transcriptomes of the two consecutive stages of cervical cancer progression corresponding to different phenotypic states—pre-invasive cancer and early invasive cancer—RNA sequencing was performed on a panel of samples comprising HPV(+) cervical intraepithelial neoplasia grade 3 (designated here as CIN_#, including carcinoma in situ, *n* = 4) and early invasive squamous cell carcinoma at FIGO IA1-IIB stages (designated as CR_#, *n* = 7), plus morphologically normal epithelium (*n* = 1) (Table 1).

Taking into consideration the fact that early invasive carcinoma may neighbor intraepithelial lesions and other constraints associated with the analysis of native biopsy specimens, and in order to evaluate representativeness of CIN/CR samples and the resultant cDNA libraries selected for the search of differentially expressed genes (DEGs) involved in (lymph-)angiogenesis and EMT regulation, we first compared the expression patterns of genes encoding structural proteins that determine cell type identity. It is well established that the development of HPV-associated cervical neoplasia is accompanied by progressive loss of expression of epithelial phenotype markers defining stratified squamous epithelium, i.e., directing the processes of epidermal (keratinocyte) differentiation, keratinization and cornification, and the establishment of apical/basal cell polarity. These markers are represented by proteins composing specific cell structures, such as epithelial-specific cell–cell and cell–extracellular matrix (ECM) contacts, cytoskeletal and ECM components, including the cornified cell envelope constituents (Figure 2). A common feature shared by mesenchymal cells is their ability to form focal contacts and various types of membrane protrusions; in addition, mesenchymal morphology is typically characterized by a distinct pattern of cytoskeletal proteins, secreted components of ECM, and matrix degrading enzymes. In contrast to structural markers of epithelial tissue organization, the expression of mesenchymal phenotype markers can become elevated as malignant cells acquire an invasive phenotype [15,16].

Heatmaps showing expression patterns of genes of described functional categories in our panel of CIN and CR specimens can be seen in Figure 3 (see Appendix A for the complete gene list; group A and B genes). As expected, morphologically normal HPV(-) epithelium displayed the largest difference between the expression of epithelial and mesenchymal markers (with strong expression of the former and weak expression of the latter). A relative decline in the expression of epidermal/epithelial organization markers with concomitant upregulation of a suite of mesenchymal markers could be seen when comparing early invasive carcinoma versus high-grade CIN, which was taken as a criterion of tissue sample representativeness and correspondence between the derived transcriptome libraries and histopathological diagnosis. Sample CIN_1 was then excluded from DEGs analysis, as its expression level of mesenchymal markers turned out to be significantly below those of normal epithelium.

Next, considering the main task of this study consisting in identification of genes involved in regulation of (lymph-)angiogenesis and EMT during “intraepithelial-to-invasive” switch, we examined transcriptome profiles of CIN and CR samples for common marker genes that, according to Gene Ontology, are responsible for angiogenesis-, (lymph-)angiogenesis-, and EMT-related Biological Processes (GO-BPs) (Figure 4). Based on GO Molecular functions, these common genes, positive and negative regulators (as well as DEGs described hereafter), were categorized into the following functional groups. (1) Genes encoding growth factors, cytokines, chemokines, and other cytokine-like proteins interacting with signaling receptors; (2) genes encoding cytokine and growth factor receptors; (3) genes encoding transcription factors; (4) genes encoding membrane proteins or protein components of cell junctions and ECM constituents that mediate interactions between cells and their extracellular environment; and (5) other members of cellular pathways with signal transducer activity (Appendix A; group C genes).

The pattern of proximities/dissimilarities of the resulted gene expression profiles is depicted in Figure 5. CIN and CR samples formed two clusters separated from normal epithelium. Samples CIN_1 and CR_4 were dropped as they did not group with any of these clusters. Then, a comparison of the transcriptomic profiles was performed on all CR samples against all CIN samples; however, this did not result in identification of significant DEGs, which was likely be due to the proximity of CR_2, CR_3, and CR_6 samples to CIN cluster. Therefore, the whole-transcriptome screening of DEGs was continued between the most “distant” samples that fall into two clusters corresponding with the two different phenotypic states: early-cancer invasive (CR_1, CR_5, CR_7) versus pre-invasive (CIN_2, CIN_3, CIN_4) (outlined in red and black, Figure 5A). After the whole-transcriptome comparison, the remaining samples diverged into the same two clusters corresponding to different states (Figure 5B).

The genes with the base 2 logarithmic fold change value |logFC| larger than 1.0 and false discovery adjusted *p*-value (FDR) < 0.05 were referred to as significant genes differentially expressed (DEGs) between CIN and CR clusters. With these cut-off criteria, a total of 201 DEGs were identified, of which 49 genes were upregulated and 152 downregulated in CR samples relative to CIN (Figure 6 and Appendix A, which contains the complete list of DEGs arranged according to their FDR values in ascending order).

Among these 201 DEGs, there were a considerable number of genes that, according to literature and GO database, have been known to participate in (lymph-)angiogenesis, EMT, reorganization of cytoskeleton, ECM remodeling, cell migration, and promotion or suppression of invasion and metastasis in various cancer types (in Appendix A, genes participating in angiogenesis, lymphangiogenesis, EMT, and epithelial/epidermal differentiation are highlighted in corresponding colors, in accordance with GO annotation).

Figure 7 and Figure 8 show heatmaps of the expression changes of the DEGs in the derived CIN/CR sample panel. Importantly, many of them have been known to act as pleiotropic regulators contributing to both angiogenesis and EMT (Appendix A). In addition, we noticed high presence of genes involved in inflammatory immune response (including interferon-mediated inflammation) and belonging to the system of self/non-self DNA/RNA recognition and DNA-damage response (colored in green in Appendix A). Among cytokine genes and genes with growth factor-related function displaying elevated mRNA expression, chemokines with well-established proangiogenic and EMT-promoting activity (*CXCL9, CXCL10*, and *CX3CL1*; Figure 7 and Figure 8) were found upregulated in CR samples. Among genes encoding cytokine and growth factor receptors and upregulated in CR, the expression of *EPHB2*, a member of the ephrin receptor tyrosine kinases family (subclass B), whose essential involvement in various aspects of malignant growth has only recently been recognized due to high complexity of mechanisms controlled by them [17], was markedly increased. *EPHB2* has been reported to promote in vitro angiogenesis in head and neck cancer [18] and induce EMT in cervical cancer cells [19]. Interestingly, in contrast to *EPHB2*, the expression of *EPHA3*, a member of another subclass of the ephrin receptors, was significantly decreased in invasive carcinoma samples (Figure 8). Despite some controversy over its precise role, *EPHA3* is indeed regarded as a negative regulator of EMT and lymphogenic metastasis in particular types of epithelial cancers [20,21]. Four genes encoding proteins with DNA-binding transcription factor-related activity (*E2F7, HMGB2, MYBL2*, and *NSD2*) and two genes encoding chromatin modifying enzymes (*DNMT1* and *UHRF1*) showed elevated transcript levels in CR samples, of which *MYBL2* and *NSD2* were confirmed to function in pathways directly implicated in EMT in HPV(+) cervical cancer cells in in vitro studies [22,23,24]. Importantly, some of these proangiogenic and/or pro-EMT DNA-interacting factors identified as DEGs (e.g., *MYBL2*/B-Myb and *DNMT1*) are known to be directly incorporated into viral mechanisms of carcinogenesis or to be the components of DNA-sensing and DNA-damage response pathways, thereby serving as mediators of inflammation and viral infection recognition (e.g., *PML*), along with other cytosolic factors (e.g., *STMN1* and *LAPM3*) also found to be upregulated in CR samples (Figure 8) [25,26,27].

As mentioned above, a greater number of DEGs (152 of 201) turned out to be downregulated in CR compared to CIN, with many known tumor suppressor genes (*AR, PGR, FAT4, PTCH1, TGFBR3, HOPX, SASH1, CRYAB, CFTR,* and *CLCA4*) or candidate tumor suppressor genes (*PTPRM, PTPRU, SLIT2, LMO7, NEGR1, AIF1L, DEPTOR,* and *RHCG*) found among them. Several of them exert their tumor-suppressing activity via enhancing homophilic cell–cell adhesion (*MPZL3, FAT4, GJB2, LMO7, NEGR1, PCDH18, CRYAB*, and *SERPINB1*), maintaining epithelial integrity, or modulating the WNT-, TGFBR-, SHH-, and Hippo signaling pathways and retinoic acid metabolism, exerting thereby anti-EMT and anti-metastatic effect. For some of these genes (*DPP4, EDN3, SLIT2, DKK1, SFRP4,* and *PTCH1*), the mechanisms of HPV-driven epigenetic silencing have been uncovered [28,29,30,31,32]. Among the genes downregulated in invasive CR samples, we surprisingly found *EMP1, EREG*, *DCN*, and *DPP4*, the genes commonly associated with positive regulation of angiogenesis and motility. Nevertheless, several studies reported on decreased expression of these genes in some types of epithelial cancers. For instance, reduced level of *DPP4* (CD26) expression/activity was detected in HPV(+) cervical cancer cell lines compared to non-cancerous cell lines [33]. By integrated bioinformatics analysis of cervical cancer datasets from the Gene Expression Omnibus (GEO), *DPP4* and *SFRP4* were identified among the 8 top downregulated hypermethylated tumor suppressor genes [34]. Decorin (*DCN*), a crucial player in physiological angiogenesis, shows an obvious inhibitory function in angiogenesis within the tumor microenvironment [35]; for example, microinvasive carcinoma of the cervix displayed lower *DCN* expression than premalignant lesions, as evidenced by immunohistochemistry [36]. There have also been a lot of contradictory data about the role of *EMP1*, epithelial membrane protein 1 (summarized in [37]); nevertheless, anti-migratory function of *EMP1* in nasopharyngeal cancer cells [38] and its downregulation upon oral dysplasia progression to oral squamous cell carcinoma [39] have been documented; in addition, the mechanism for anti-lymphangiogenic effect of *EMP1* through downregulation of *VEGFC* expression was proposed [38]. Epiregulin (*EREG*) considered as a hub gene in squamous cervical carcinoma and other gynecological tumors [40] has been reported to undergo silencing in certain types of carcinoma [41]. A plausible explanation for this discrepancy may lie in the multifaceted role of genes like *EMP1* and *EREG*: in addition to angiogenesis, they are involved in a variety of immune reactions, including regulation of T cell chemotaxis and activation and inflammation, as well as in ECM–cell interactions and ECM remodeling, and epithelial differentiation. A similar reason may underlie decreased expression of *ALDH1A1* and *CD24*, cancer stem cell-like markers commonly associated with more aggressive features and EMT phenotype, seen in CR samples (Figure 8). However, there is data showing that ALDH^high^ expressing cancer stem cell-enriched spheroid culture cells derived from cervical and head and neck cancers are more sensitive to CD8 (+) cytotoxic T lymphocyte-mediated killing compared to ALDH^low^ cells [42]. Furthermore, a decreased level of *ALDH1A1* has been revealed in neoplastic cells of non-small-cell lung carcinoma compared to non-neoplastic epithelial cells by immunohistochemistry, with this decrease tended to be correlated with stronger proliferative activity [43]. Concerning the relationship between *CD24* expression and the EMT features, heterogeneity of cervical cancer cells has been recently reported: only cells having CD44 (+) CD24 (-) phenotype exhibited mesenchymal traits and attained invasive, migratory and spreading abilities [44]. Consistent with this report, while all CIN and CR samples in our dataset expressed equally high levels of *CD44*, *CD24* demonstrated a clear tendency toward a decreased transcript level in CR versus CIN (Appendix A).

Curiously enough, comparison of transcriptomes of pre-invasive and early invasive cervical cancer revealed virtually no significant “classical” promoters of angiogenesis/lymphangiogenesis conventionally used to describe “angiogenic/lymphangiogenic switch”, such as members of VEGF/VEGFR-, FGF/FGFR-, and PDGF/PDGFR families; angiopoietins; and proangiogenic (*ETS* and *HIF*) and pro-lymphangiogenic (*PROX1*, *SOX18*, and *FOXC2*) transcription factors. The same was seen with EMT-promoting factors: although many of the identified DEGs function in the WNT-, TGFBR-, SHH-, and Hippo pathways, there were no typical “master regulator” transcription factors (members of *SNAI, TWIST, ZEB,* and *FOX* families) or signaling receptors (*HGFR/MET* and *NOTCH*) among them. Of mentioned gene families, *VEGFA* appeared to be the only gene expressed at relatively higher level in CR samples, though its fold-change value did not achieve significant FDR-adjusted *p* < 0.05 (FDR = 0.1254, Appendix A). Supposing this result be due to small sample size, evaluation of the relative expression of selected key genes associated with (lymph-)angiogenesis and EMT regulation was attempted using real-time PCR and a separate panel of samples, which comprised, additionally to CIN3/carcinoma in situ (*n* = 12) and early invasive cancer (stage IA, *n* = 10), benign HPV(+) cervical intraepithelial lesions (CIN1, *n* = 8) and morphologically normal cervical epithelium (control, *n* = 10). Specifically, mRNA expression levels of genes encoding pro-(lymph-)angiogenic and pro-EMT growth factors *VEGFC, VEGFD, PlGF,* and *HGF*; tyrosine-kinase receptors *VEGFR3* and *MET/HGFR*; transcription factors *ETS1, PROX1*, and *SLUG/SNAI2*; and anti-angiogenic cell adhesion molecule and ECM component *THBS1*, were measured relative to mean expression of four reference genes (see Methods). Generally, as the disease stage increased (from CIN1 to CIN3 and invasive cancer stage IA), the relative expression levels varied in a wider range of values, with several genes (*VEGFC, PlGF, HGF, VEGFR3*, and *MET*) exhibiting a tendency towards higher expression in carcinoma in situ and invasive carcinoma stage IA relative to the control, but the differences between the cancer and the control groups, as well as between CIN3/CIS and stage IA cancer proved statistically insignificant (Figure 9A; the exception, *SLUG*, displayed significantly higher mRNA level in HPV+/CIN1 and CIN3 samples compared to the control, *p* < 0.05, *U*-test). Such pattern of expression variation was retained at the protein level as revealed by western blot (Figure 9B), consistent with the RNA-Seq results (Appendix A). Nevertheless, in spite of absence of statistically significant differences between pre-invasive CIN and early invasive CR, some of these genes demonstrated coordinated changes in their mRNA expression as revealed by correlation analysis: specifically, *VEGFC/VEGFR3* and *HGF/MET* pathway members exhibited moderate to strong correlation (*p* < 0.01, Figure 10), suggesting that these genes may cooperatively contribute to the initial stage of invasive cervical cancer progression and that there were some unaccounted factors underlying variation in their expression in CIN/CR.

In summary, the results of RNA-Seq allowed us to speculate that, during the onset of invasive growth, promotion of angiogenesis, EMT, and migration of cervical cancer cells might be realized mainly through downregulation of genes antagonizing these processes (potential tumor suppressors), rather than upregulation of “classical” proangiogenic or pro-EMT genes. In addition to protein-coding genes, we also noticed sharp changes in the expression levels of two long non-coding RNAs (lncRNA), *RMST* (downregulated in CR samples), and *LINC01305* (overexpressed in CR samples) (Figure 8). Of note, *RMST* was proposed as a putative regulator of NOTCH/WNT signaling pathways [45] and a tumor suppressor lncRNA; for example, in triple-negative breast carcinoma, *RMST* has been shown to restrain the invasion and migration abilities [46]. On the contrary, *LINC01305* has been demonstrated to directly potentiate EMT in cervical cancer cells and was identified as the most notably overexpressed lncRNA in cervical carcinoma [47,48].

Concomitantly with the observed changes in the expression of angiogenesis-/EMT-regulating genes, a number of epidermal differentiation markers (namely, *CSTA, CSTB, CRNN, DMKN, DSG1, EPGN, KLKs 7, 8, 13, KRT13, PPL, PPP1R3C, SBSN, SCEL, SPINK5, SPINK7, SPRR2A, SPRR3, TMPRSS11B, TMPRSS11E*, and *TGM1*) were, as anticipated, significantly repressed in CR samples compared to CIN (Appendix A). Several of these genes belong to a gene cassette (1q21) named the epidermal differentiation complex [49]; other genes located in this region (as, for example, *IVL, LOR, RPTN, FLG2, LCE3D, LCE3E,* and *S100* genes) were also suppressed in CR samples, albeit not achieving the FDR-significance level of 0.05 (indicated by corresponding color, Appendix A).

Regarding a phenomenon of “cadherin switch”, a key phenotypic attribute of EMT, we did not observe “classic” E-to-N cadherin switch in the RNA-Seq profiles of CIN/CR samples due to a comparably high expression of E-cadherin (*CDH1*); however, CR samples displayed a notably higher expression level of P-cadherin (FDR < 0.1, Appendix A), resulting in the higher P/E-cadherin ratio (6-fold, on average). Considering the existing experimental evidence that, upon tumoral progression, P- and E-cadherins are able to cooperatively maintain partial-EMT (a hybrid-EMT phenotype) and facilitate collective cell migration and invasion [50,51], providing that the expression of E-cadherin is preserved or slightly decreases and the expression of P-cadherin concurrently increases, our results may give additional evidence for these notions.

### 2.2. Pathway Analysis

To further characterize which regulatory mechanisms are associated with the transition of high-grade intraepithelial neoplasia (or carcinoma in situ) to early invasive cervical carcinoma and, accordingly, which signaling pathways are implicated in a shift towards a different phenotypic state (that is, pre-invasive-to-invasive), functional enrichment analysis was performed on significant DEGs. The GO analysis suggested significant enrichment in 30 “biological processes” (GO-BPs) (Figure 11). Remarkably, GO-BPs related to blood vessel development and morphogenesis and tube morphogenesis, in general, appeared to be among the top most significant terms (according to their FDR-values), along with the groups of terms that define epithelial/epidermal differentiation (Figure 11A). Many of the GO-BPs identified could be regarded as tightly interacting during the early phases of invasive cervical cancer progression, which is likely due to a relatively broad spectrum of functions for the genes they involve; particularly, Figure 11B shows a big cluster of interconnected terms related to vasculogenesis and angiogenesis. Other significant groups of terms, such as, for example, response to estradiol, simply illustrate relation to the female reproductive system or the processes commonly affected in cancer (such as cell proliferation, response to stress, etc.). Detailed list of genes and GO terms ranked by FDR can be seen in Appendix A.

Furthermore, five KEGG Pathways (“DNA replication”, “Influenza A”, “Axon guidance”, “Pyrimidine metabolism”, and “Renin secretion”) were also significantly enriched (Table 2). Genes encoding enzymes of pyrimidine metabolism are recognized as immune checkpoint mediators involved in regulation of T cell activation and inflammation (e.g., *CD73/NT5E* and *CD203c/ENPP3*), on the one hand, and as angiogenesis regulators, on the other hand. Members of the endothelin system in the “Renin secretion” pathway (*EDN* and *EDNRA*) are intrinsically implicated in angiogenesis, lymphangiogenesis, and vascular functions. All identified components of the “Axon guidance” pathway are known to act as crucial cytoskeletal remodelers; sensors of the ECM mechanic properties; and regulators, either positive or negative, of directed cell migration [53], representing a functional bridge between inflammation, angiogenesis, EMT, and invasion. For instance, members of the ephrin receptor (*EPH*) family have been shown to mediate crosstalk between TNFα- and TGFβ-signaling pathways [54]. In addition, *EPHB, SLIT*, and *UNC5* were found to play a role in lymphangiogenesis [55]. The genes from the “Influenza A” pathway are associated with interferon (IFN) type I and type II signaling, which corroborates viral nature of cervical oncogenesis. At the same time, under chronic inflammatory conditions, these genes have been implicated in complex interactions between tumor cells and their microenvironment, including endothelial cells, thereby potentiating angiogenesis and guiding migration of tumor cells towards the developing vasculature [56,57,58]. And the last significant pathway comprised, quite expectedly, components of the DNA replication machinery. Overall, despite being part of quite specific mechanisms, the genes identified in the KEGG analysis are endowed with a much broader spectrum of functions in tumorigenesis as recent studies have revealed. Taken together, KEGG enrichment analysis suggests the signaling pathways collectively sustaining the processes of vasculature development, directed migration of various types of cells (tumor, endothelial, and immune), inflamed microenvironment, and immune suppression to be critically important for transition to the invasive stage of cervical cancer progression.

To find out what factors and mechanisms may lie behind alterations of gene expression patterns and signaling pathways occurring as pre-invasive neoplasia transitions to invasive cervical carcinoma, significantly up- and downregulated genes were subjected to Signaling Pathway Enrichment using Experimental Datasets (SPEED) tool [59]. As shown in Figure 12, SPEED enrichment analysis resulted in five signaling pathways with an adjusted *p*-value < 0.05: “Toll-like receptor (TLR)”, “Mitogen-activated protein kinase (MAPK)”, “Tumor necrosis factor (TNF) α”, “Interleukin (IL) 1”, and “WNT”. Genes identified as targets of the TLR pathway suggest the antiviral response or DNA-damage response to be the prime cause of transcriptome perturbations observed at the initial stage of invasion. Simultaneously this implies the connection between invasion and inflammatory microenvironment, hypoxia, angiogenesis, epithelial (de-)differentiation, and EMT, as these genes directly participate in all these processes. Selectivity of gene expression changes of mediators of inflammation involved in TLR-pathway is of noteworthy mention: for example, while the expression of *CXCL10*, a proinflammatory chemokine, was increased in invasive CR samples, the expression of *IL-18* (“IFNγ-inducing factor”) was, conversely, decreased. The coupling of TLR signaling pathway to the CIN3-to-cancer transition was also noted by Yi et al. [60]. Another finding that the genes significantly up/downregulated in the early-stage invasive carcinoma were significantly enriched for components of the TNFα- and IL1- signaling pathways suggests the inducing role of the inflamed microenvironment. Of note, those of TNFα- or IL1- downstream targets, that were found significantly upregulated in CR samples (i.e., *CX3CL1, CXCL10, MMP3, TYMP,* and *GPR68*), are known to act as regulators of angiogenesis and, prominently, lymphangiogenesis, as well as EMT, cell migration, and invasion. At the same time, *IL1RN* (inhibitor of IL1 signaling), *DKK1* (inhibitor of WNT signaling), and proto-cadherin *PCDH18* (inhibitor of EMT and invasion) showed decreased expression.

As expected, the MAP kinase cascade (ERK/MEK/MAPK), a well-recognized target for HPV oncogenes [61,62], was also identified upstream of invasion-associated transcriptome perturbations (Figure 12). The genes involved in this pathway represent quite specific modulators of angiogenesis, invasion and immune response. For example, *TNFRSF21* (DR6, death receptor 6), showing increased expression in CR samples, is considered a marker of tumor-associated vascular endothelium and a mediator of tumor cell migration [63]. *LMO7*, downregulated in CR samples, has been reported to stabilize homotypic adherens junctions, maintain epithelial integrity and inhibit TGFβ signaling [64,65,66]. The WNT signaling pathway, with three specific genes (*HMGB2, CD24*, and *DKK1*, all being widely discussed in connection with the regulation of EMT and tumor cell invasion) found to be perturbed in early-stage invasive cervical cancer, has also been proved a major HPV target, along with the p53 and Rb pathways [67]. Concerning antiviral response and immune regulation, an overlap of the pathway enrichment results from KEGG with the results from SPEED appears to be of interest, suggesting that the establishment of a persistent inflammatory microenvironment might be a relevant mechanism upstream of angiogenesis, lymphangiogenesis, and EMT induction in relation with the initial stages of invasive cervical cancer growth.

Finally, to provide additional support for the significance of the identified signaling pathways for promotion of invasive behavior and angiogenesis of early-stage cervical cancer, a total of 201 up- and downregulated DEGs were submitted to The Search Tool for the Retrieval of Interacting Genes/Proteins (STRING) database to construct a Protein–Protein Interaction (PPI) network, which enables gaining information for predicted physical and functional interactions between genes and their protein products [68]. In sum, STRING analysis revealed close interrelation of genes differently expressed between early-stage invasive cervical cancer and its noninvasive precursor, with a total of 170 nodes mapped in the PPI network (shown in Figure 13, with the exception of disconnected nodes). Especially noticeable were gene connections that joined the molecular components of epithelial cell inflammatory response to pathogenic self- or foreign DNA or DNA-damage response (including interferon-stimulated genes (ISGs)), as well as the components of inflammation-associated angiogenesis and EMT, including pro- or anti-inflammatory and proangiogenic cytokines/chemokines and pyrimidine metabolism enzymes.

### 2.3. Flow Cytometry for FLT4/VEGFR3, MET/HGFR and SLUG/SNAI2 Expression

In analyzing the above transcriptome and signaling pathway data that revealed implication of angiogenesis-related processes in the transition between pre-invasive neoplasia and early-invasive cancer, the following considerations were made. (1) Early-stage cervical cancer was enriched for the immune-related genes and immune-controlling pathways, including ISGs, cytokines, mediators of inflammation, immunoregulatory factors, and mechanisms of pathogen recognition; (2) by contrast, many core regulators of (lymph-)angiogenesis and EMT displayed no significant changes in invasive cancer; (3) high level of variation in expression of certain well-known drivers of (lymph-)angiogenesis/EMT resulted in the lack of statistical significance when analyzing the entire panel, pointing to heterogeneity of native tumor samples; and (4) a range of a mesenchymal phenotype markers exhibited no statistically significant upregulation (with some of them showing an apparent trend toward increased expression) and a range of epithelial markers maintained original expression level (despite a certain decreasing trend) in a fraction of CR specimens. We reasoned that these results could mirror (at least partially) the influence of the following factors (or processes). (1) The impact of the immune microenvironment; (2) epithelial–mesenchymal plasticity (EMP) [69]; (3) existence of post-transcriptional regulation resulting in that changes in genes’ expression are chiefly realized at the level of their protein products [69]; (4) cellular heterogeneity, particularly if the expression of a certain (lymph-)angiogenesis/EMT marker is associated with a specific phenotype of cells, that constitute a minor subpopulation of a tumor; and, finally (5) the probability that changes in the expression of well-recognized (lymph-)angiogenesis and EMT markers may occur earlier than morphologically distinguishable invasion is activated (that is, at CIN2-3/carcinoma in situ) should also not be discarded.

Following these assumptions, we made an attempt to take into account possible contribution of the above factors by analyzing three “classical” regulators of (lymph-)angiogenesis and EMT—cell surface receptors VEGFR3/FLT4 and HGFR/MET and a transcription factor SNAI2/SLUG—and performing enzymatic dissociation approach, antibody staining and flow cytometry on a separate set of samples, comprising morphologically normal cervical epithelium (control, *n* = 5), benign HPV(+) cervical intraepithelial lesions (benign/CIN1, *n* = 7), CIN2-3/cancer in situ (*n* = 12) and early invasive cervical cancer (IA stage, *n* = 7).

The gating strategy applied for discrimination between cell populations of interest is shown in Figure 14. To differentiate between immune and non-immune components of cell suspensions, anti-CD45 antibody was used. The tumor cell-enriched population was identified by anti-pan-cytokeratin (CK) and anti-EpCAM (CD326) staining. Co-expression pattern of pan-CK and EpCAM epithelial markers was considered as one of the signs of epithelial–mesenchymal plasticity (EMP) or partial-EMT, based on published data on head and neck cancer that retention of cytokeratin expression combined with gradual loss (or keeping at lower level) of EpCAM expression was the evidence of the processes constituting tumor EMP [16,69]. Accordingly, enumeration of HGFR/MET(+), VEGFR3/FLT4 (+) and SLUG/SNAI2 (+) cells was carried out among CK (+) EpCAM (+), CK (+) EpCAM (-) and CK (-) EpCAM (+) populations gated from CD45 (-) cells (Figure 14).

The total amount of CK (+) cells in a CD45-negative subset gradually increased as the malignancy progressed from CIN1 to carcinoma in situ, and in invasive carcinoma of stage IA, it became significantly higher than in controls (*p* < 0.05, *U*-test), while the amount of EpCAM (+) cells was elevated in CIN1 samples (*p* < 0.05), showing no significant changes at higher CIN grade and cancer (Figure 15). When analyzing the co-expression pattern for these two markers inside the CD45-negative subset, we watched CK (+) EpCAM (-) population underwent the largest increase with disease progression from CIN to cancer (with the peak at stage IA), with CK (+) EpCAM (+) cell counts being also elevated. Conversely, the percentage of CK (-) EpCAM (+) cells, being raised in benign lesions, was drastically decreased with an increase in neoplasia severity (*p* < 0.05, U-test). Correspondingly, the ratio CK (+)/EpCAM (+) cells (in %) displayed a tendency toward increase in invasive cervical cancer samples as compared the control. Inside the CK (+) subset, the proportion of EpCAM (+) cells, being first increased (in CIN1), tended to decrease in high-grade CIN and cancer, indicating that the development of invasive cervical carcinoma may be accompanied with expansion of cells that lose their EpCAM expression, whilst preserve cytokeratins, which could be considered as a sign of the EMP program, rather than full EMT program (Figure 15) [16].

As follows from Figure 16A, FLT4- and MET-expressing cells were very rare among CD45-negative cell subset of the control samples and HPV(+) benign lesions/CIN1, while in high-grade lesions or carcinoma in situ we detected a noticeable increase in their numbers (*p* < 0.05, *U*-test), which became even higher in invasive stage IA cervical cancer; however, the difference between CIN3/cancer in situ and cancer stage IA was statistically confirmed only for MET (*p* < 0.05, *U*-test). A yet larger increase demonstrated the relative median fluorescence intensity increase (ΔMFI) suggesting that not only the proportion of MET/FLT4 bearing cells increase upon cancer development, but the intensity of cells’ expression (surface density) of these markers as well (Figure 16B). The frequencies of MET (+) and FLT4 (+) cells were correlated (*r* = 0.828, *p* < 0.01). Regarding SLUG, a distinct pattern of changes was observed in that the relative amount of SLUG (+) cells increased at as early as HPV(+) benign lesions/CIN1, with the peak at CIN3/cancer in situ (*p* < 0.05, *U*-test), and then, at stage IA cancer, clearly tended to decrease, but remained statistically significantly higher than the control values, consistent with the real-time PCR results (Figure 9A) and RNA-Seq. No correlation was found between the relative abundances of MET (+)/FLT4 (+) and SLUG (+) cells within CD45 (-) gate.

Next, FLT (+), MET (+), and SLUG (+) cells were profiled for CK and EpCAM. As seen in Figure 16C, the expression of FLT4 and MET markers appeared to be associated predominantly with CK (+) cells, with the abundance of CK (+) EpCAM (-) phenotype being sharply elevated in cancer stage IA samples. As a result, the CK (+) EpCAM (-) subpopulation accounted for a majority of FLT4/MET expressing cells in stage IA. The proportion of CK (+) EpCAM (+) cells displayed no significant changes; however, it was on average higher in MET (+) subset, than in FLT4 (+). CK (-) EpCAM (+) phenotype was detectable among FLT4/MET-positive cell fraction in pre-cancerous lesions at a relatively low frequency. Unlike FLT4 and MET, the expression of SLUG turned out to be associated primarily with EpCAM (+) population, while the frequency of CK (+) EpCAM (-) phenotype among SLUG (+) cells was low in all sample groups analyzed.

If we compare the percentages of specifically CD45 (-) CK (+) EpCAM (-) FLT4 (+), CD45 (-) CK (+) EpCAM (-) MET (+) or CD45 (-) EpCAM (+) SLUG (+) cells, then the difference between pre-invasive neoplasia and early invasive cancer becomes even more pronounced (Appendix A). Concerning distinctive pattern of SLUG expression (Figure 16A,C), the study by Puram et al. [16] carried out on oral cavity tumors with the use of single-cell transcriptomics is worth attention, since they found SLUG to be the only “classical” EMT-driving transcription factor upregulated in malignant cells exhibiting an EMT-like phenotype and, presumably, the earliest trigger of the partial-EMT program.

When evaluating the proportion of FLT4/MET/SLUG expressing cells within total CD45(-) CK(+) and CD45 (-) EpCAM (+) subsets (gates P6 and P7, respectively, Figure 14B), it turned out that it was cervical cancer stage IA where the majority of CD45 (-) EpCAM (+) cells and a substantial part of CD45 (-) CK (+) cells were FLT4 or MET positive (Appendix A), suggesting a role for these receptors in driving EMT program and sustaining aggressive invasive behavior of epithelial tumor cells and associated lymphangiogenesis. SLUG-positivity was detected in a considerable proportion of CD45 (-) EpCAM (+) cells in cervical cancer precursor lesions, but a notably lower number of CD45 (-) CK (+) cells stained positively for SLUG in all stages examined (Appendix A). In this regard, it is noteworthy to mention the results reported by Cui et al. who found SLUG to exhibit tumor-suppressor activity in cervical carcinoma cells [76]. Therefore, the observed nonlinear dynamics of SLUG expression changes may hypothetically imply that its upregulation is an initial and necessary step for EMT induction and angiogenesis at a pre-cancer stage, but in subsequent invasive progression, its down-modulation is a prerequisite for accelerated tumor cell proliferation, possibly specific to cervical cancer.

We also noted that cervical cancer samples tested here varied substantially in the relative amount of intraepithelial CD45 (+) immune component including tumor-infiltrating lymphocyte (TILs) counts (gate P5, Figure 14). The TILs counts in patient samples were on average higher than in control samples, this difference being statistically significant in HPV+/CIN1 and CIN2-3/cancer in situ groups (*p* < 0.05 *U*-test, Figure 17). Following this, we compared the percent of TILs with the expression of selected markers on cells negative for CD45 and found samples of pre-invasive carcinoma and stage IA carcinoma displayed moderately strong negative correlation between the percentage of TILs and the abundance of FLT4 (+) or MET (+) tumor cells (FLT4: *r* = −0.672 at *p* = 0.017, R^2 = 45.19, MET: *r* = −0.516 at *p* = 0.085, R^2 = 26.65, ANOVA); by contrast, the frequency of SLUG(+) cells was positively correlated with the percentage of TILs (*r* = 0.660 at *p* = 0.019, R^2 = 43.61, ANOVA). In addition, when the samples of CIN3/cancer in situ and stage IA were divided into two groups with the proportion of lymphocytes amounting to <5% or >5% based on empirical distribution function for TILs frequencies, it occurred that the expression levels of FLT4, MET, and SLUG differed between these two groups at *p* < 0.05 (*U*-test) (Figure 17), suggesting a role for the local immune profile in cervical neoplasia progression.

Taken together, the comparative analysis of expression changes of three “classical” lymphangiogenesis and EMT regulators (VEGFR3, MET, and SLUG), accompanying the transition from intraepithelial neoplastic lesions to early invasive cervical cancer, performed at the cellular level revealed that (1) the expression pattern of VEGFR3, MET, and SLUG is specific for cell populations that display a particular epithelial phenotype resembling the features of EMP-like phenotype; (2) alterations in the immune composition may play a pivotal role in establishing invasive ability of cervical cancer cells; and (3) activation of (lymph-)angiogenesis/EMT regulators’ expression can take place in cervical epithelial cells at the stage of intraepithelial tumor development, supporting initial assumptions.

## 3. Discussion

Notwithstanding that the majority of cervical cancer cases are diagnosed at a preclinical stage having virtually 100% curability, its advanced metastatic stages are regarded to be a fatal disease and one of the leading causes of cancer death among females. The mechanisms of cervical cancer cell invasion and metastasis—as a fundamental scientific and clinical challenge—are increasingly becoming the focus of genome-scale molecular profiling investigations, including transcriptome analysis, with diverse experimental designs and approaches being used for identifying driving molecular alterations and, accordingly, potentially effective therapeutic targets. One such approach consists in comparing molecular profiles of invasive cervical cancer specimens of different clinical stages and searching for associations with clinical features (such as metastases, response to therapy, recurrence-free or overall survival). Most comprehensively, this approach has been implemented in a recent large-scale study of cervical cancer by The Cancer Genome Atlas consortium [77]. Using RNA sequencing and other NGS applications, the authors were able to characterize three molecular clusters of cervical cancer, which span, though incompletely, its main histological subtypes. Intriguingly, besides expression of keratin gene family members, other ectoderm development genes, and some significantly mutated genes, these clusters were associated with distinct EMT scores (defined as the difference between the averaged expression levels of mesenchymal and epithelial genes) and with patients’ overall survival, emphasizing the importance of profiling for EMT-related genes in elaborating cervical cancer treatment strategies.

A somewhat different approach to elucidating important molecular pathways taken in the present work is based on a consideration that a managing role in tumor expansion may belong to mechanisms that are switched on with the beginning of the earliest phase of invasive tumor growth, i.e., in superficially invasive carcinoma, antecedent to initiation of metastatic spread [4]. The ability to detect such “molecular switch” during evolution of high-grade squamous intraepithelial lesion/cervical carcinoma in situ to microcarcinoma has been recently demonstrated in two studies [78,79]. Although this was done on the example of a remarkable change in intensity and distribution of immunohistochemical expression of single markers (PD1L and Ki67, in one work, and podoplanin, in another work), it is reasonable to assume this switch may engage a large number of genes responsible for regulation of angiogenesis, EMT, and invasion, which can be discovered by means of transcriptome profiling of pre-invasive and early invasive carcinoma. Such RNA-Seq-based transcriptome profiling has been conducted for microinvasive breast cancer (namely, emerging invasive fingers/EIF and carcinoma in situ/CIS), in comparison with normal breast epithelium and invasive breast cancer [80], which led to identification of 20 upregulated and 446 downregulated DEGs between the epithelial cell fractions of noninvasive CIS and microinvasive EIF implicated in cancer-related pathways, actin cytoskeleton and focal adhesion. This approach certainly has some limitations pertaining primarily to diagnostic (histological) errors and sampling complexities, but they can be counterbalanced with parallel analysis of additional molecular markers mirroring structural reorganization of epithelial tissue during invasion [78].

In our choice of a set of epidermal differentiation markers to monitor representativeness of cervical tissue specimens, we built on the results of several experiments on global gene expression carried out in a model of HPV16-infected keratinocytes [15,81,82]. According to RNA-Seq data from [15], HPV16 infection disturbs the activity of more than 3000 genes, a considerable proportion of which is related to epithelial barrier function, including genes responsible for keratinization and cell–cell contacts. At the same time, infection itself does not activate EMT genes (or may even suppress *HGF, FOXC2*, and *VEGFR* genes) and is not able to induce an invasive phenotype formation. Similar conclusion was made by Kang and co-authors based on cDNA microarray analysis of transcriptome alterations caused by early productive HPV16 infection in 3D organotypic raft cultures of cervical keratinocytes [81]. In the study by Yang et al. [82], the in vitro model of immortalized female human oral keratinocytes stably expressing the HPV16 *E6/E7* and RNA-Seq were employed to explore how viral oncogenes dysregulate host–cell pathways associated with cellular organization, cell contacts, cell motility, and ECM composition. As for CIN and invasive cervical carcinoma, which natural development is coupled with the loss of a productive HPV life cycle, there is also experimental evidence delineating progressive impairment of epithelial/epidermal differentiation processes upon cervical cancer growth [8,9,83,84]. In our work, the samples of pre-invasive neoplasia and early invasive carcinoma displaying marked differences in the expression of epidermal differentiation markers were selected for further analysis. Moreover, when comparing our data with the results of transcriptome profiling for cervical cancer of IB-IIA-IIB stages *versus* norm [9], a considerable overlap in the spectrum of downregulated markers of cornification, desquamation and the epidermis functions (namely, *KRTs, SPRRs, TGMs, SBSN, SPINKs, CRNN, KRTDAP, SCEL, DSG1, TMPRSS 11B/11E, PPP1R3C*, and KLK8/13) could be seen. Similarly, our findings agree with the results of bioinformatics analyses of transcriptomic data carried out on invasive cervical cancer and normal tissues from GEO database [83,84], particularly concerning the reduced expression of *SPINK7, PPL, PPP1R3C, SPRR1A/1B/3, DSG1/2, CRNN*, and *CSTB*. Altogether, all changes in the “(lymph-)angiogenesis”, “EMT”, and “invasion” gene expression profiles accompanying transition of pre-invasive lesions to the early invasive cervical carcinoma were observed simultaneously with a dramatic reduction in the abundance of transcripts typical of normal keratinocyte phenotype, which was used to support that we were dealing with tissue samples that were close to each other in terms of cervical cancer staging, but different in terms of phenotype.

Global epigenetic silencing of tumor suppressor genes is believed to be a primary strategy for cervical cancer to progress [34,85,86]. Our results add to this knowledge in that this strategy might be utilized at transition from intraepithelial stage of cancer development to the stage of invasive tumor growth. Of 201 DEGs identified, 152 genes were downregulated in early invasive carcinoma relative to its immediate precursor, high-grade CIN, many of which are well-established tumor suppressor genes. For most DEGs, the main function consists in regulation of cell–cell and cell–ECM communications, so that these genes are commonly considered as regulators of EMT and angiogenesis or lymphangiogenesis. Of these DEGs, we turned our attention to members of three of the four known cell guidance molecule families (the netrins and their receptors, *SLITs* and their receptors, ephrins and their receptors, and semaphorins and their (co-)receptors plexins and neuropilins), which are increasingly implicated in carcinogenesis [87,88]. By mediating interactions between epithelial, endothelial, and immune cells and governing the processes of tissue remodeling, cell contractility, and migration, these molecular factors are critically involved in neovascularization, epithelial–mesenchymal transformation, invasion, and inflammation; however, their role in cervical cancer is much less studied than in other cancer types. In our search for genes differently expressed between early invasive carcinoma and pre-invasive cervical neoplasms, significant downregulation of *UNC5C* (netrin receptor), *SLIT2/3* (ligands for the ROBO receptors), and *EPHA3* (ephrin receptor type A), and upregulation of *EPHB2* (ephrin receptor type B) were identified in invasive cervical cancer (FDR < 0.05). In addition, KEGG analysis revealed enrichment of “Axon guidance” signaling pathway. While *UNC5* expression in cervical neoplastic tissue is still poorly studied, *SLIT2* and *SLIT3* have been described to be frequently inactivated (due to DNA methylation or deletion) in invasive cervical cancer, with the frequency showing gradual increase as stage progresses [29,89]. In spite of the fact that this inactivation can be detected in earlier, premalignant, lesions, the most significant increase in percentage of promoter hypermethylation of both *SLIT2* and *SLIT3* occurs, according to Narayan et al., when high-grade intraepithelial neoplasia progresses into invasive cervical carcinoma [29], our expression data being in line with these findings.

The importance of the ephrin/EPH-mediated signaling for HPV(+) cervical cancer progression was demonstrated in the work by Kori et al. [84], who performed analysis on five independent datasets for invasive cervical carcinoma and normal tissue from GEO and identified reporter receptors, which led to cancer progression-associated transcriptome changes. Of these 18 reporter receptors, three were the ephrin receptors type A and B, including *EPHB2* (found among the upregulated DEGs in our RNA-Seq dataset). As for *EPHA3* (downregulated in our dataset), there is few literature data available on its expression in cervical cancer, while in other epithelial cancers (e.g., head and neck cancer [90], colorectal cancer [20], esophageal cancer [21], and clear-cell renal cell carcinoma [91]), *EPHA3* has been reported to be a tumor suppressor; an inhibitor of angiogenesis, EMT, and cell migration and invasion; and a potential predictor for tumor spread, being frequently downregulated or inactivated due to promoter hypermethylation or mutation. We also gave consideration to *EPHA7*, another member of the ephrin type A receptor family: despite not corresponding FDR-adjusted p-value cut-off (FDR = 0.0599 > 0.05), its expression showed a 3-fold decrease in early invasive cancer compared to CIN3/cancer in situ. Although little is known about the role of *EPHA7* in cervical cancer pathogenesis, its frequent downregulation brought about by various epigenetic mechanisms and coupled with tumor progression was determined in other epithelial malignancies, as, for example, in prostate cancer [92], colorectal cancer [93], and esophageal squamous cell carcinoma [94]. Given that the ephrin receptors are the largest known subfamily of receptor tyrosine kinases with diverse functions and modalities of action, including ligand-independent activities (such as, for example, mechanoresponsive signaling) [53,95], identification of the two members of this family among the significant genes differently expressed at the onset of invasive cervical cancer progression seems to be an important result that requires further investigation.

The presence of EPH receptors among the DEGs identified prompted us to look at another large family of signaling molecules, that can integrate the processes of angiogenesis, lymphangiogenesis, and EMT during tumor progression—the semaphorins (SEMAs). Although none of SEMA genes reached statistical significance (FDR > 0.05) and none of them were considered DEGs, the expression of several SEMAs displayed a clear tendency toward being altered in the early invasive cancer stage (with |logFC| = 1.5−4.8 and a *p*-value that trended toward significance, Appendix A), with the directionality of these alterations being in agreement with the literature data about pro-/anti-angiogenic or pro-/anti-metastatic activity of SEMAs in different epithelial tumors. In particular, the level of *SEMA4B, SEMA4C* and *SEMA4F* transcripts was higher, while the level of *SEMA3B* and *SEMA3E* was lower in early-stage invasive cancer samples as compared with CIN3/cancer in situ. Indeed, while class 4 semaphorins are characterized mostly as pro-tumorigenic and proangiogenic SEMAs, class 3 semaphorins, with a few exceptions, are regarded as endogenous inhibitors of (lymph-)angiogenesis [96,97]. From the TCGA pan-cancer data analysis, Zhang et al. inferred that *SEMA3B* and *SEMA3E* were predominantly downregulated in cancer [98]. *SEMA3E*, altered by logFC of 4.8 in our study (Appendix A), has been previously found to function as an inhibitor of tumor development, angiogenesis and metastasis (discussed in [97]). Of mentioned here *SEMA4B, SEMA4C* and *SEMA4F* genes, *SEMA4C* has been explored in more detail in the context of epithelial cancer progression (whereas literature data on *SEMA4B* and *4F* remain scarce). *In vitro*, in various experimental cell models of cervical cancer, *SEMA4C* was able to potentiate tumor lymphangiogenesis and lymphatic metastasis [99], modulate TGFβ-mediated EMT [100] and promote EMT-mediated cisplatin resistance [101]. *SEMA4F* has been described to correlate, if overexpressed, with increased motility of cells, increased invasion and disease aggressiveness in prostate cancer [102]. Similar to *SEMA4C, SEMA7A* (which showed a 3-fold increase, albeit insignificant, in our RNA-seq dataset) has been reported to have an effect on cells of other epithelial cancer types, for instance, to stimulate migration of oral tongue squamous cell carcinoma by regulating TGFβ-induced EMT signaling [103] and to orchestrate macrophage-mediated lymphatic vessel remodeling to drive metastasis in breast cancer [104]. Not only semaphorins’ expression, but the expression of some of their receptors (plexins *PLXNA1, PLXNA4,* and *PLXNB3*) as well showed a certain tendency to be altered during the switch between intraepithelial development and invasive growth (Appendix A). One can anticipate more significant differences with expanding the amount of in situ and early-stage invasive (microinvasive) carcinoma specimens; nevertheless, the already obtained results allow us to hypothesize that the earliest stages of invasive cervical cancer progression might be associated with highly specific changes in the expression pattern of the semaphorin family members and their receptors, dependent on their specific functions. Although the research on the role of this class of molecular regulators in cervical cancer pathobiology is lagging behind in comparison with other tumor types, it is apparent that they play crucial roles—along with the other cell guidance factors mentioned above—in activation of invasion and further tumor progression.

Another important finding from the produced RNA-Seq and pathway analyses is that the earliest stages of invasive cervical cancer appeared to be linked to changes in the expression of inflammatory genes and innate immune response pathways suggesting that the inflammatory microenvironment might be a provoking factor for cervical cancer cell invasion, angiogenesis and lymphangiogenesis. Generally, HPV-driven carcinogenesis represents a commonly acknowledged example of how chronic inflammation caused by a persistent infection becomes a driving force for tumor development [105]. It is also well accepted that an inflammatory response can contribute to the development of a vascular network within the tumor [106], including lymphatic vascular network [107], and EMT [108]. Besides, the members of the interferon-mediated antiviral response (interferon-stimulated genes, ISG) may facilitate tumor-associated lymphangiogenesis and enhance lymphatic metastasis capabilities [109]. However, as argued by Aguilar-Cazares et al., while the relationship between inflammation and angiogenesis in the advanced stages of cancer is indeed supported by numerous studies, far fewer reports describe the association of these processes in the early stages of cancer [110]. Therefore, gaining research evidence of this relationship is still relevant, especially for various borderline conditions, such as, for example, the transition from carcinoma in situ to early invasive carcinoma. Such research has been recently conducted by Chen et al. based on the hepatitis C virus-induced liver cancer dataset encompassing transcriptomic profiles of its progressive stages (cirrhosis, low-grade dysplastic, high-grade dysplastic liver samples, very early cancer, early cancer samples, and advanced and very advanced cancer) [4]. According to Chen et al., inflammation represents a critical transition state between the norm and the very early cancer stage, with angiogenesis considered as a key regulator and a hallmark of this inflammation-to-cancer-transition; while enrichment of cell adhesion regulatory factors, in particular, ECM–receptor interaction regulators, is an indicator of the early cancer stage [4]. Our RNA-Seq results also corroborate that, in virus-associated cervical cancer, inflammation and other mechanisms of innate immune response to viral infection could trigger invasive behavior, angiogenesis and EMT upon transition from intraepithelial neoplasia to invasive carcinoma. Engagement of the pyrimidine metabolism genes, an emerging class of immune checkpoints (Table 2), together with proinflammatory factors, supports the idea that inflammation can give rise to the immunosuppressive tumor microenvironment formation, which, in turn, further contributes to angiogenesis and migratory behavior of tumor cells at the transition from pre-cancer to cancer [111].

In addition to investigating transcriptome profiles of pre-invasive neoplasia and early invasive cancer of the cervix, in the second part of this study we made an attempt to figure out whether it is possible to survey the observed correlations—such as changes in the expression of “(lymph-)angiogenesis/EMT” genes co-occurring with epithelial phenotype marker expression alterations and local immune deviations as pre-invasive cancer transitions to invasive carcinoma—at the cellular level. We applied a flow cytometric approach to evaluate the distribution of VEGFR3/FLT4 and HGFR/MET receptors and SLUG/SNAI2 transcription factor (as representing pivotal regulators of the processes being studied) within populations of cells isolated from native cervical cancer tissues, which helped to take into account possible impact of post-transcriptional regulation [69] and cell heterogeneity, including the presence of immune cell infiltrates. Flow cytometry of single-cell suspensions prepared by enzymatic dissociation of surgical tumor tissue specimens is now being extensively used not only for phenotyping and quantitative analysis of tumor-infiltrating immune cells, but also for studying the expression of angiogenesis-related biomarkers, epithelial–mesenchymal plasticity and self-renewal abilities in tumor cells, tumor-associated fibroblasts, endothelial cells, and other cell types composing tumor stroma. Concerning cervical cancer and particularly its earliest forms, previous research has been concentrated mainly on the assessment of tumor-infiltrating immune cells in clearly invasive stages (starting with stage IB and higher). Conversely, in the current work, we were focused on the CD45-negative, non-immune cell component obtained from preclinical cervical cancer samples.

Assessment of EpCAM and CK markers chosen to define the tumor cell enriched population revealed retention of epithelial traits in cells derived from stage IA cervical cancer, on the one hand, and redistribution of EpCAM and CK relative expression as CIN evolves to invasive carcinoma, on the other hand (Figure 15). Interestingly, in a study by Pan et al. [112] on circulating tumor cells (CTCs) from patients with cervical cancer of I–II stages, a high overall prevalence of CTCs with EpCAM(+)CK8(+) epithelial phenotype (especially in stage I) and a presence of CTCs exhibiting a mixed phenotype were described, with the incidence of the latter being increased with the stage progression and with bigger depth of stromal invasion and pelvic lymph nodes involvement, illustrating the epithelial-mesenchymal phenotypic plasticity of cervical cancer cells capable of invasion and metastasis at early clinical stages. According to our results, specific changes in the protein expression of VEGFR3/FLT4, HGFR/MET, and SLUG/SNAI2 were associated with the EMP-like phenotype of tumor cells. These changes were already detectable in high-grade intraepithelial lesions and became further enhanced in the early invasive cervical carcinoma (except for SLUG/SNAI2) (Figure 16), with MET showing a statistically significant difference between pre-invasive and early invasive neoplasms.

Our literature search, as well as published meta-analyses and reviews [113,114], gives grounds for stating that the involvement of FLT4, MET, and SLUG in cervical cancer pathogenesis has been poorly investigated as contrasted with other tumor types. At the same time, the attractiveness of MET/HGFR and FLT4/VEGFR3 as prognostic biomarkers and targets for anti-metastatic, anti-angiogenic and, since recently, immune therapy for a variety of epithelial malignancies has come from numerous observations that they synergistically contribute to tumor growth (via auto- and paracrine mechanisms), (lymph-)angiogenesis, and tumor cell invasion/migration, and participate in formation of immune tolerance and suppression via immune checkpoint mechanisms [115,116,117]. In this study, we found VEGFR3, MET and SLUG expression on cells derived from CIN3/cancer in situ/cancer stage IA tissue samples to be associated with the proportion of tumor infiltrating lymphocytes, with VEGFR3 and MET showing negative correlation and SLUG showing positive correlation with %TILs (Figure 17). Indeed, as demonstrated in previous studies, HPV-positive tumors could be heavily infiltrated with leukocytes (both lymphocytes, and myeloid cells), with the microenvironment they contribute to characterized as inflammatory, wound-healing (proangiogenic) and immunosuppressive [75,85,111,118,119,120,121]. At the same time, many other studies have documented the link between higher immune score or greater lymphocyte infiltration (especially CD8) with better overall survival in patients with advanced cervical cancer, and with regressing disease—in patients with high-grade intraepithelial lesions, supporting an important role of lymphocyte-mediated immunity for controlling the disease [111,121,122]. Although the factors underlying quantitative and qualitative characteristics of the immune (particularly, lymphocytic) infiltrate present in cervical tumors are the subject of special investigations and were not the task of the current research, nevertheless stratification of the FLT4, MET, and SLUG expression values in accordance with the amount of tumor-infiltrating lymphocytes resulted in more homogenous groups. Inverse relationship we observed between the relative abundance of TILs and the proportion of VEGFR3/FLT4 (+) and HGFR/MET (+) cells may imply the immune system actively participates in controlling (lymph-)angiogenesis and EMT during transition from intraepithelial neoplasia to early invasive/microinvasive carcinoma. Overall, these findings accentuate a more and more extensively discussed relationship between the processes of (lymph-)angiogenesis, EMT and invasion, on the one hand, and the immune inflammatory mechanisms and the functions of TILs, on the other hand, during virus-associated carcinogenesis.

The strength of the present study, in our opinion, is that we used native primary tissue samples from our clinical practice to perform comparative transcriptome analysis. At the same time, we are fully aware of its limitations. First of all, this is low number of samples. The second part of limitations relates to accuracy of tissue sampling hampered by microscopic sizes of CIN and carcinoma of 0-IA stages, co-localization of cervical epithelial lesions at different neoplastic stages, and the risk of capturing variable amounts of a stromal component. We made an effort to counterbalance these drawbacks by maximally precise tissue sampling and by matching samples’ histology with their RNA-Seq profiles of epidermal/epithelial differentiation marker expression, based on which a fraction of CIN/CR specimens were excluded from further DEG analysis of “(lymph-)angiogenesis” and “EMT” profiles (these excluded specimens nevertheless need further investigation as part of an extended sample panel). Finally, one more shortcoming concerns the statistical data filtration strictness: given small size of the resultant transcriptomic datasets, in addition to analysis of significant DEGs, we drew attention to genes that failed to reach FDR cut-off, but were altered by |logFC| >1 (at *p* < 0.01) between the two phenotypic states, as they could contain important regulators and potentially informative markers. In total, there were up to 700 such genes, with many of them indeed known to be implicated in (lymph-)angiogenesis, EMT, regulation of cell morphology, migration ability, cell–ECM interactions, and ECM remodeling (Appendix A, the gene names are color coded by known GO functions). Concerning individual genes, some published research reporting their crucial role in cervical cancer pathobiology can be found in PubMed. It should be noted that, as the disease stages being analyzed in the present work usually do not constitute a diagnostic or therapeutic problem and have standardized and quite effective treatment methods, we were not aimed at creating a specific prognostic or diagnostic signature. Identification of such signatures requires a particularly rigorous approach and a much broader patient coverage. However, those differences in molecular profiles between the two consecutive phenotypic states (pre-invasive versus early invasive carcinoma of the cervix) we were able to determine in this descriptive study open new possibilities for more detailed future research.

## 4. Materials and Methods

### 4.1. Patients and Samples

Tissue samples were obtained from patients with HPV(+) benign cervical lesions, cervical intraepithelial neoplasia (CIN) of grade 1–3 (CIN3 comprised carcinoma in situ cases) and early-stage invasive squamous cell carcinoma of the cervix at FIGO stages IA-II (including microinvasive cancer with invasion of the stroma up to 3 mm in depth and 7 mm of extension) during a colposcopy-directed biopsy or surgery. Patients underwent treatment in the Republican Oncological Dispensary; in each case, the diagnosis was histologically verified by histopathologists, who also inspected the prevalence of cancerous cells in malignant tissue with minimal inclusion of underlying loose connective tissue stroma to ensure the biopsy sampling accuracy. The presence of the high-risk carcinogenic HPV infection was confirmed. HPV-negative morphologically normal cervical epithelium was taken as a control. The research was approved by the Committee on Medical Ethics at the Institute of Medicine of Petrozavodsk State University and the Ministry of Healthcare and Social Development of the Republic of Karelia (protocol No.39, Approval date: 30 May 2017), and was done in accordance with the Declaration of Helsinki and good clinical practice guidelines. The diagnosis was based on comprehensive physical examination, extended colposcopy findings, cytology and histopathology tests, in full compliance with the approved standards for the diagnosis and treatment of patients with gynecological malignancies. All women engaged in this this study were informed and gave voluntary written consent.

### 4.2. RNA Sequencing (RNA-Seq)

Cervical tissue samples were placed in IntactRNA stabilization reagent at +4 °C immediately after excision (during surgery). Total RNA was isolated using TriZOL reagent (Invitrogen, Carlsbad, CA, USA). The quality and quantity of isolated RNA were assessed based on 28S:18S rRNA ratio using Fragment Analyzer automated system (Advanced Analytical, Santa Clara, CA, USA). cDNA libraries were constructed using TruSeq Stranded Total RNA (with RiboZero, Illumina, San Diego, CA, USA) kit, reverse transcriptase SuperScript III (Invitrogen, Carlsbad, CA, USA), and AMPure XP Beads (Beckman, Brea, CA, USA). The adaptors-ligated fragments were loaded onto the flow cell using MiSeq v3 sequencing kit; 75 bp end-reads were generated on the MiSeq platform (Illumina, San Diego, CA, USA). Raw reads were filtered (sequence quality control was done with the FastQC tool); then, the filtered reads were mapped to the reference human genome (GRCh38/p13, NCBI) using STAR aligner. Quality statistics of reads are shown in Appendix A. HTSeq package was used to assess the relative abundance of transcripts which was calculated by estimating the Counts Per Million reads mapped (CPM). The generated RNA-Seq dataset has been deposited in the National Center for Biotechnology Information Gene Expression Omnibus (GEO) with accession ID GSE89361 (*f.n*.: this accession number corresponds to datasets of 2 CIN and 2 CR samples; datasets for the remaining 8 samples analyzed in this work are awaiting for obtaining accession numbers and will be provided further). DESeq2 software was applied to study gene differential expression between pre-invasive neoplastic lesions and early invasive cancer. The genes with the base 2 logarithmic fold change value |logFC| larger than 1.0 and false discovery adjusted *p*-value (FDR) <0.05 were identified as Differentially Expressed Genes (DEGs). Heat maps were used to display the expression profile of genes differentially expressed between cancer stages. Gene ontology (GO) analysis and pathway analysis were carried out on DEGs using Gene Ontology (http://www.geneontology.org) and Kyoto Encyclopedia of Genes and Genomes (KEGG) PATHWAY (http://www.genome.jp/kegg) Databases with an FDR-corrected *p*-value of <0.05 and gene count of >2 considered as the thresholds; enrichment results were visualized using ShinyGO web-tool [52]. For causal pathway analysis, Signaling Pathway Enrichment using Experimental Datasets (SPEED) algorithm was applied to DEGs [59]. In addition, in order to explore the relationship between DEGs at the protein–protein interaction (PPI) level, we used Search Tool for the Retrieval of Interacting Genes (STRING) database (https://www.string-db.org) and converted the results visually by ShinyGO software [52].

### 4.3. Real-Time Polymerase Chain Reaction (RT-PCR)

Total RNA was extracted from tissue samples with Trizol Reagent (Invitrogen). The concentration and purity of the RNA template was determined by spectrophotometry. RNA nativity was monitored by capillary gel electrophoresis using Fragment Analyzer. cDNA was synthesized from DNAse I-treated (Fermentas, Thermo Fisher Scientific Baltics, Vilnius, Lithuania) RNA (1 µg RNA per 1 reaction volume) using ProtoScript II First Strand cDNA Synthesis Kit (New England BioLabs, Ipswich, MA, USA). Amplification was performed in StepOnePlus thermal cycler (Applied Biosystems, Thermo Fisher Scientific, Carlsbad, CA, USA) using qPCRmix-HS-SYBR+HighROX reaction mix (Evrogen, Moscow, Russia). Melting curves were analyzed in each run to confirm specificity of amplification. Primer characteristics and other details concerning PCR amplification and analysis protocols used are available on demand. The levels of mRNA expression were calculated using the 2–∆Ct method (the amplification efficiency was calculated for primer pair by using standard curves). Four genes (*EEF1A1, ACTB, GAPDH*, and *RPLP0*) were taken as endogenous controls due to their proved constitutive expression in cervical tissues [123,124,125].

### 4.4. Western Blot

Frozen tissue specimens (morphologically normal cervical epithelium from HPV-negative (control) or HPV-positive women, CIN1, CIN2, CIN3 (including carcinoma in situ, or stage 0), and invasive carcinoma of stage IA) were thawed and quickly homogenized in ice-cold RIPA buffer (50 mM Tris, pH 7.5, 150 mM NaCl, 1% Triton X-100, 0.5% sodium deoxycholate, 0.1% SDS, 1 mM EDTA) containing 1 mM PMSF and 1X Halt Protease Inhibitor Cocktail (Thermo Scientific, Waltham, MA, USA), then incubated for 30 min on ice and centrifuged. Total protein concentration in supernatants was measured using Pierce BCA Protein Assay kit (Thermo Scientific, Rockford, IL, USA). Samples (25 µg of protein per well) were separated on either 8–16% or 4–20% TGX precast SDS-PAGE gels (Bio-Rad, Hercules, CA, USA) and then transferred to a nitrocellulose membrane. Membranes were blocked for a minimum of 1 h at room temperature in blocking solution (5% nonfat dry milk in 0.05% TBS–Tween 20) and, after the removal of blocking solution, probed with primary antibodies (#ab140639 for placental growth factor/PlGF, #ab83905 for vascular endothelial growth factor C/VEGFC, #ab154079 for vascular endothelial growth factor receptor 3/VEGFR3/FLT4, and #ab178395 for hepatocyte growth factor/HGF, all from Abcam, Cambridge, UK) at optimal dilutions overnight at +4°C with gentle agitation. Unbound antibody was next removed by washing the membranes three times in washing solution, each wash for 5 min, and washed membranes were then incubated with horseradish peroxidase (HRP) conjugated secondary antibody (Bio-Rad, 1:5000 dilution) for 1 h at room temperature. After the washing step (five times for 5 min), the blots were placed in ECL Western blot substrate (Pierce, Thermo Scientific, Rockford, IL, USA) and exposed to X-ray film. HeLa (human cervical adenocarcinoma cell line) Whole Cell lysate (Abcam, Cambridge, UK) was used as a positive control and anti-beta-actin mouse monoclonal antibody (#ab8224, Abcam) was used as a loading control.

### 4.5. Tissue Dissociation and Flow Cytometry

To assess the proportion (%) of cells expressing FLT4/VEGFR3 and MET/HGFR, specimens of cervical neoplastic epithelium were collected in culture medium immediately after excision, transported to the laboratory at +4 °C, and processed within 2 h after surgery. Tissue samples were washed with Hank’s Balanced Salt Solution (HBSS), dissected into small fragments (<2 mm^2^) with a sterile scalpel blade (Sigma-Aldrich, St. Louis, MO, USA) and subjected to enzymatic dissociation in HBSS containing 0.14% collagenase type I (Gibco, Bleiswijk, the Netherlands), 200 units/mL DNase I type II (Sigma Aldrich), and 5% fetal bovine serum (at +37 °C, 45 min). The resulting cell suspension was filtered through a 100 µm strainer (Miltenyi Biotec, Bergisch Gladbach, Germany) and washed in 0.02% EDTA-PBS. After erythrocyte lysis live cells were counted using trypan blue dye and then resuspended so that each probe contained no less than 10^5^ cells. The following fluorophore-conjugated monoclonal anti-human antibodies were used for phenotyping: anti-CD45-PE-Vio770 (clone: 5B1), anti-CD326/EpCAM-APC (clone: HEA-125), and anti-cytokeratin (CK)-FITC (clone: CK3-6H5) from Miltenyi Biotec (Bergisch, Gladbach, Germany), anti-FLT4/VERGR3-PE (clone: 54733), anti- MET/HGFR-PE (clone: 95106), and anti-SLUG/SNAI2-PE (clone: 666633) from R&D Systems (Minneapolis, MN, USA); respective isotype controls indicated by the manufacturers were used to monitor non-specific staining. Following incubation with antibodies against surface antigens, cells were fixed and permeabilized using buffer sets from Miltenyi Biotec and then stained for intracellular CK and SLUG. For blocking of non-specific antibody binding, FcR Blocking Reagent (Miltenyi Biotec, Bergisch Gladbach, Germany) was added to cells in accordance with the manufacturer’s instruction. Cells were acquired on MACSQuant flow cytometer and analyzed using MACSQuantify version 2.11 software (Miltenyi Biotec, Bergisch Gladbach, Germany). The gains and compensation settings were adjusted, using single-stained cells. Fluorescence Minus One (FMO) controls were used to draw gates for defining positive cell populations. In addition to the percentage of positively stained cells, the expression levels of FLT4/VEGFR3, MET/HGFR and SLUG/SNAI2 were assessed as the Median Fluorescence Intensity increase (ΔMFI) in the PE-channel relative to autofluorescence of a control (PE-unstained) cell subpopulation. No less than 20,000 cells were counted per measurement.

### 4.6. Statistical Analysis

Data analysis was performed using R software. Mann–Whitney U-test was used to evaluate the differences between the investigated groups; the difference was considered to be statistically significant at *p* < 0.05. The relationship between the parameters was investigated with the use of Pearson correlation coefficient.

## 5. Conclusions

In conclusion, despite a relatively wide range of genes displaying differential expression between pre-invasive neoplastic lesions and early invasive, true, cervical carcinoma, the results obtained indicate the existence of a certain functional gene “signature” associated with (lymph-)angiogenesis, epithelial–mesenchymal plasticity and invasion. Albeit being potentially targetable, the mechanisms governing these processes during cervical cancer development remain underinvestigated [126] and, therefore, new therapies directed against these processes should be considered with caution [127]. Perhaps the most important inference from the research presented herein is that the described differential expression profile comprises groups of genes that have received little attention until recently, but now emerging as promising targets for combined anti-metastatic therapy. First of all, these are the Eph family of receptors and other cell guidance molecules, as well as members of the WNT-, TGFBR-, SHH-, and Hippo signaling pathways, and proangiogenic cytokines. Of principal importance is the fact that many components of these signaling mechanisms play crucial roles both in (lymph-)angiogenesis and immune reactions, raising potential implications for combined immunotherapy [128]. Interconnection of expression patterns of “(lymph-)angiogenesis” and “EMT” genes and some parameters of local immunity at cervical cancer transition from its intraepithelial to invasive form seems to be another interesting finding reinforcing the potential of a combination therapy approach. With respect to metastatic or recurrent cervical cancer treatment, this conception and its rationale has been underpinned by several recent studies on multigene transcriptional and immune profiling. For example, combinations of the immune checkpoints with the components of DNA-damage repair system or EMT regulators are reviewed in [126]. Unlike most such works that deal with cervical cancer ≥IB stages, our study design was based on the notion that the differences in the molecular profiles between intraepithelial neoplasms and the very early stages of invasive cancer may represent a ‘pool’ of mechanisms and crucial factors (particularly, the angiogenesis, EMT and inflammatory response genes), that act as driving forces for subsequent tumor progression and metastasis and, thus, have the potential for being used as biomarkers or therapeutic targets [4]. Both approaches warrant further exploration.

## Figures and Tables

**Figure 1 ijms-21-06515-f001:**
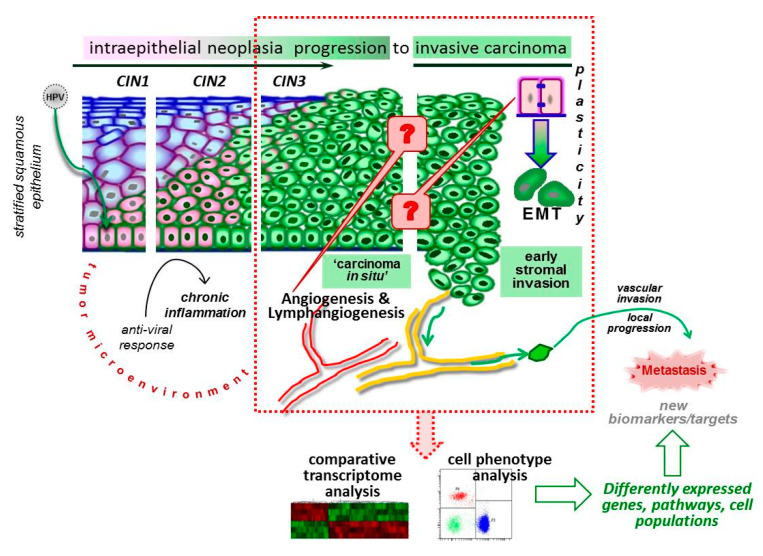
A schematic model depicting early stages of cervical cancer development and progression. Intraepithelial-to microinvasive cancer transition is outlined with a dotted box. Question marks designate the processes of angiogenesis, lymphangiogenesis, and EMT as being the main subject of the study.

**Figure 2 ijms-21-06515-f002:**
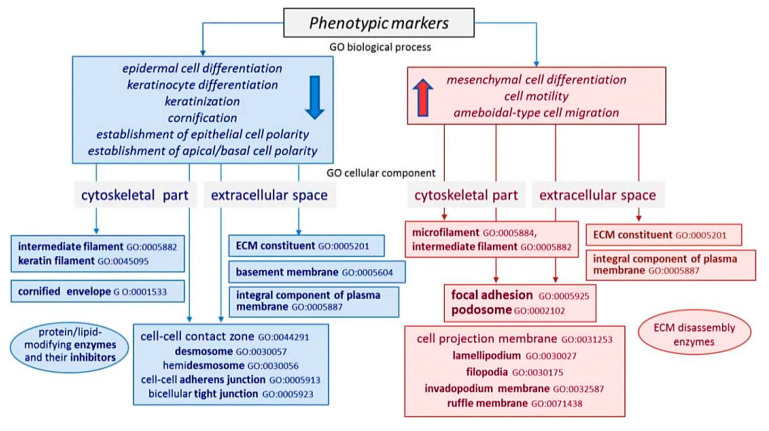
Functional categories of genes and their products responsible for epithelial (shown in blue) or mesenchymal (shown in red) cell morphology (cell shape) according to Gene Ontology (GO) terms. Red and blue arrows indicate up- and downregulation of the processes related to mesenchymal and epithelial differentiation, respectively.

**Figure 3 ijms-21-06515-f003:**
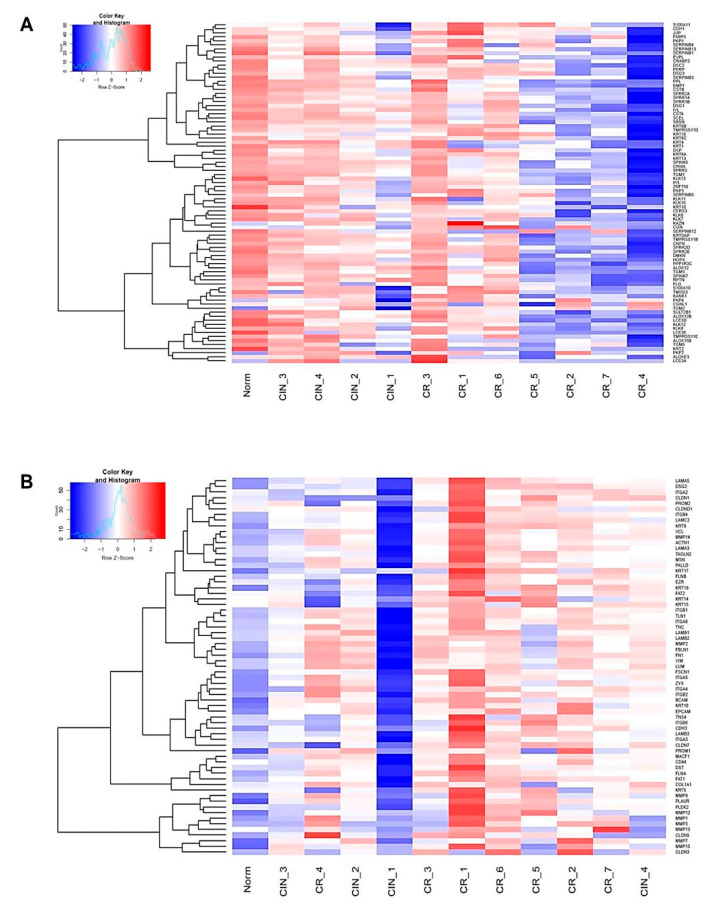
Clustered heatmap illustrating gene expression patterns of epithelial (**A**) and mesenchymal (**B**) phenotype markers in pre-invasive CIN and early-stage invasive CR samples tested.

**Figure 4 ijms-21-06515-f004:**
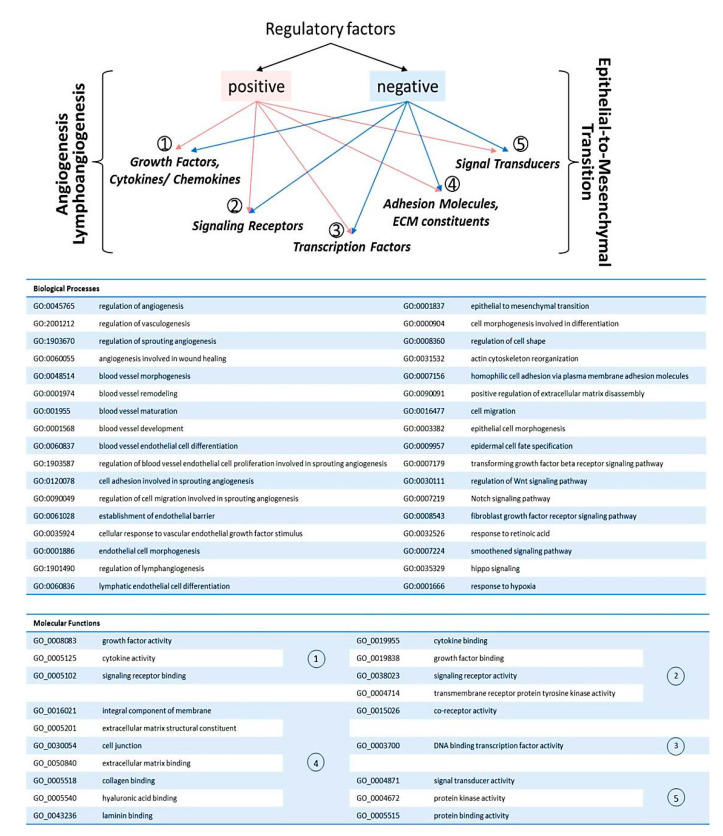
Functional categories of genes regulating angiogenesis, lymphangiogenesis, and EMT.

**Figure 5 ijms-21-06515-f005:**
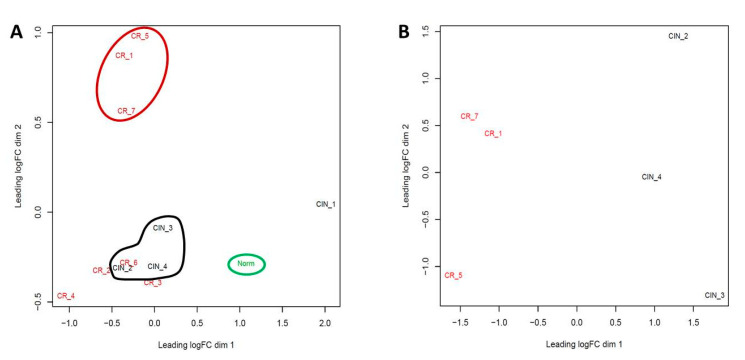
The pattern of similarities between RNA-Seq samples. (**A**) Multidimensional scaling (MDS) plot showing the distance between normal (outlined in green), CIN and CR samples according to their expression profiles of genes associated with the (lymph-)angiogenesis and EMT regulation. Two clusters consisting of 3 CIN and 3 CR samples (outlined in black and red, respectively) were selected for further comparative transcriptome analysis. (**B**) MDS plot for the whole-transcriptome RNA-Seq profiles of selected CIN and CR samples, confirming that they segregate in two clusters corresponding to the two different states (i.e., early-cancer invasive and pre-invasive cancer).

**Figure 6 ijms-21-06515-f006:**
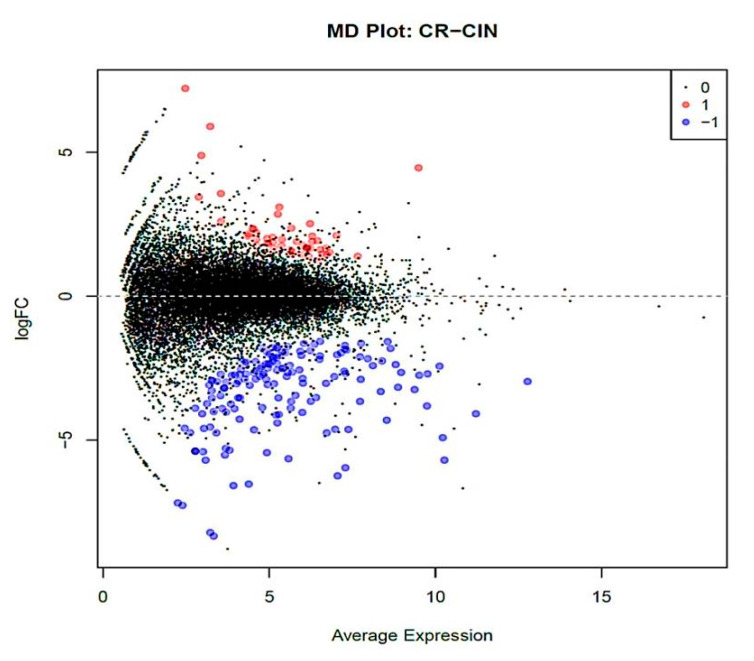
The mean-difference (MD) plot showing the average expression against logFC identified by RNA-Seq in invasive cervical cancer samples (CR_1, CR_5, CR_7) compared with pre-invasive intraepithelial neoplasia (CIN_2, CIN_3, CIN_4) samples. Significantly up- and downregulated DEGs are highlighted in red and blue, respectively.

**Figure 7 ijms-21-06515-f007:**
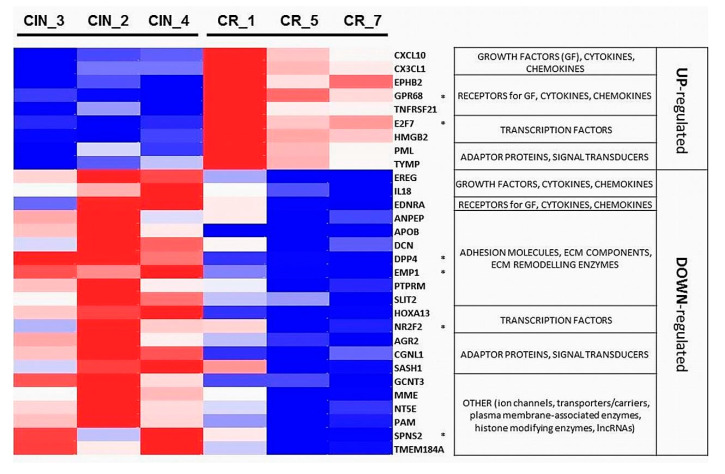
Heatmap showing the overview of the gene expression profile for regulators of angiogenesis and lymphangiogenesis (listed in the order according to the designated functional groups). Up- and down-regulated genes are shown in red and blue, respectively. Asterisks (*) point to genes implicated in lymphangiogenesis.

**Figure 8 ijms-21-06515-f008:**
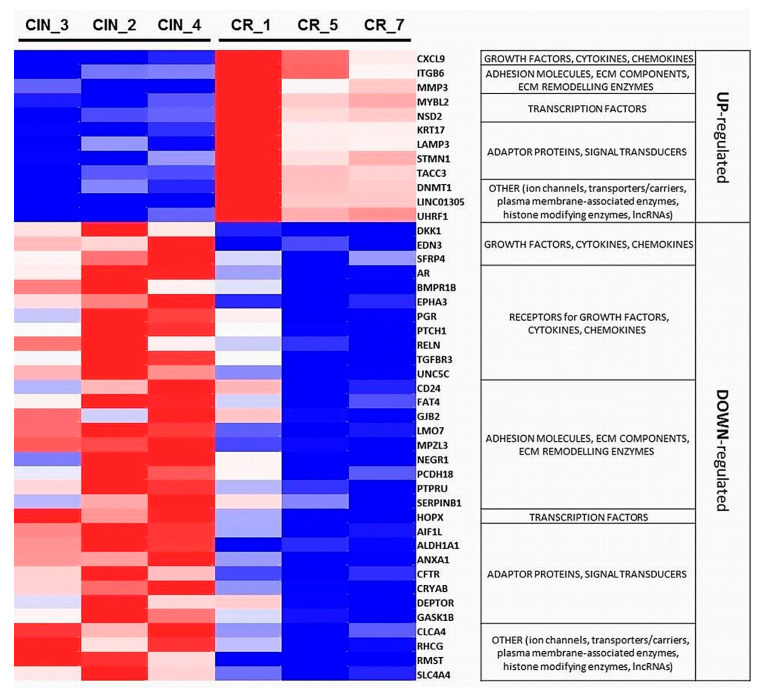
Heatmap depicting expression profile for EMT regulators (listed in the order according to the designated functional groups). Up- and down-regulated genes are shown in red and blue, respectively.

**Figure 9 ijms-21-06515-f009:**
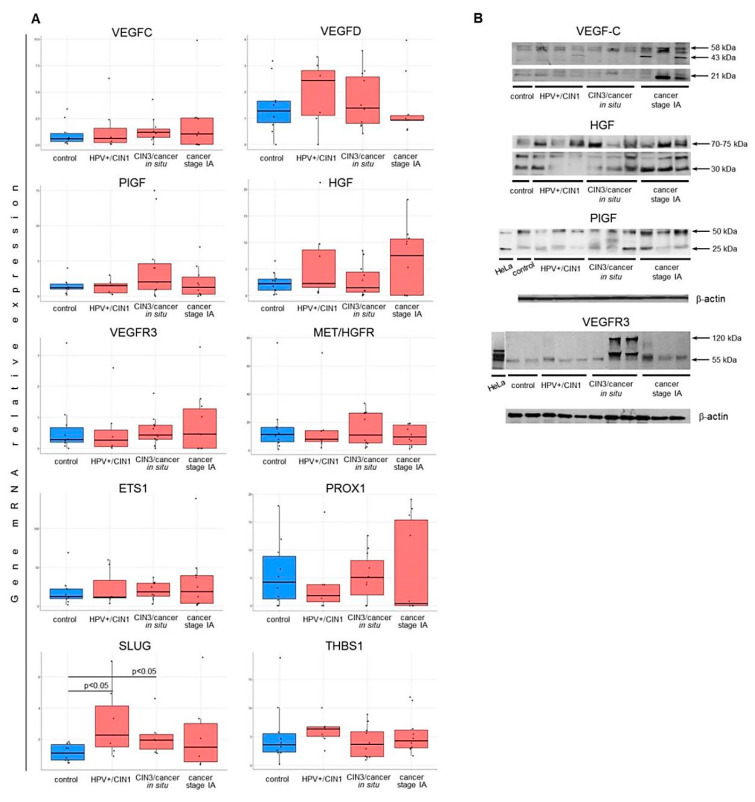
The expression of selected genes encoding biomarkers of (lymph-)angiogenesis and EMT. (**A**) Real-time PCR results showing relative expression levels of selected (lymph-)angiogenesis/EMT regulator genes normalized to 4 reference genes (*EEF1A1, RPLP0, ACTB*, and *GAPDH*) and multiplied by 10,000, in patient samples of cervical intraepithelial neoplasia (CIN), cervical cancer, and normal control. In the boxplots, individual values are shown as dots, the lower and upper hinges correspond to the first and third quartiles, and the middle line corresponds to the median; the upper/lower whisker extends from the hinge to the largest/smallest value no further than 1.5*IQR (the interquartile range). Significant differences between the patients and the controls are designated as *p* < 0.05 (Wilcoxon Mann–Whitney *U*-test); (**B**) Western blot results for protein expression of selected (lymph-)angiogenesis/EMT regulators in patient samples. *VEGFC*—vascular endothelial growth factor C, *VEGFD*—vascular endothelial growth factor D, *PlGF*—placental growth factor, *HGF*—hepatocyte growth factor, *VEGFR3*—vascular endothelial growth factor receptor 3, *MET/HGFR*—hepatocyte growth factor receptor, *ETS1*—E26 transformation specific sequence 1 transcription factor, *PROX1*—Prospero homeobox protein 1, *SLUG*—Snail Family Transcriptional Repressor 2, *THBS1*—thrombospondin 1.

**Figure 10 ijms-21-06515-f010:**
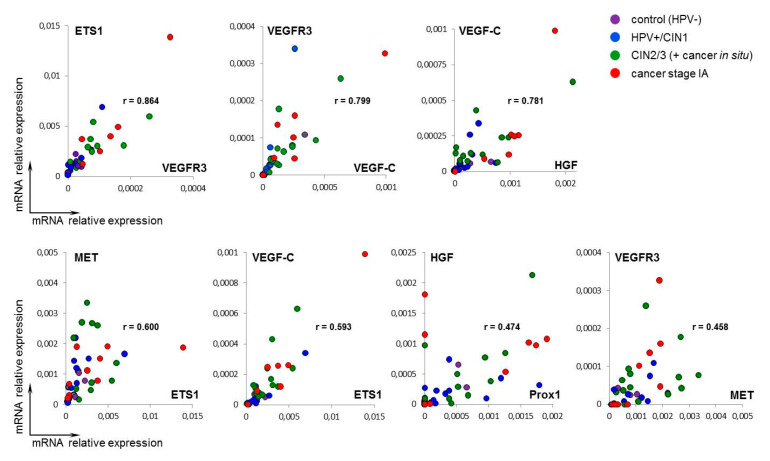
Scatterplots illustrating pairwise correlations (*p* < 0.01) between mRNA expression levels of several selected genes, well-recognized as being essential regulators of (lymph-)angiogenesis and EMT, at early steps of cervical cancer development; *r*—Pearson’s correlation coefficient.

**Figure 11 ijms-21-06515-f011:**
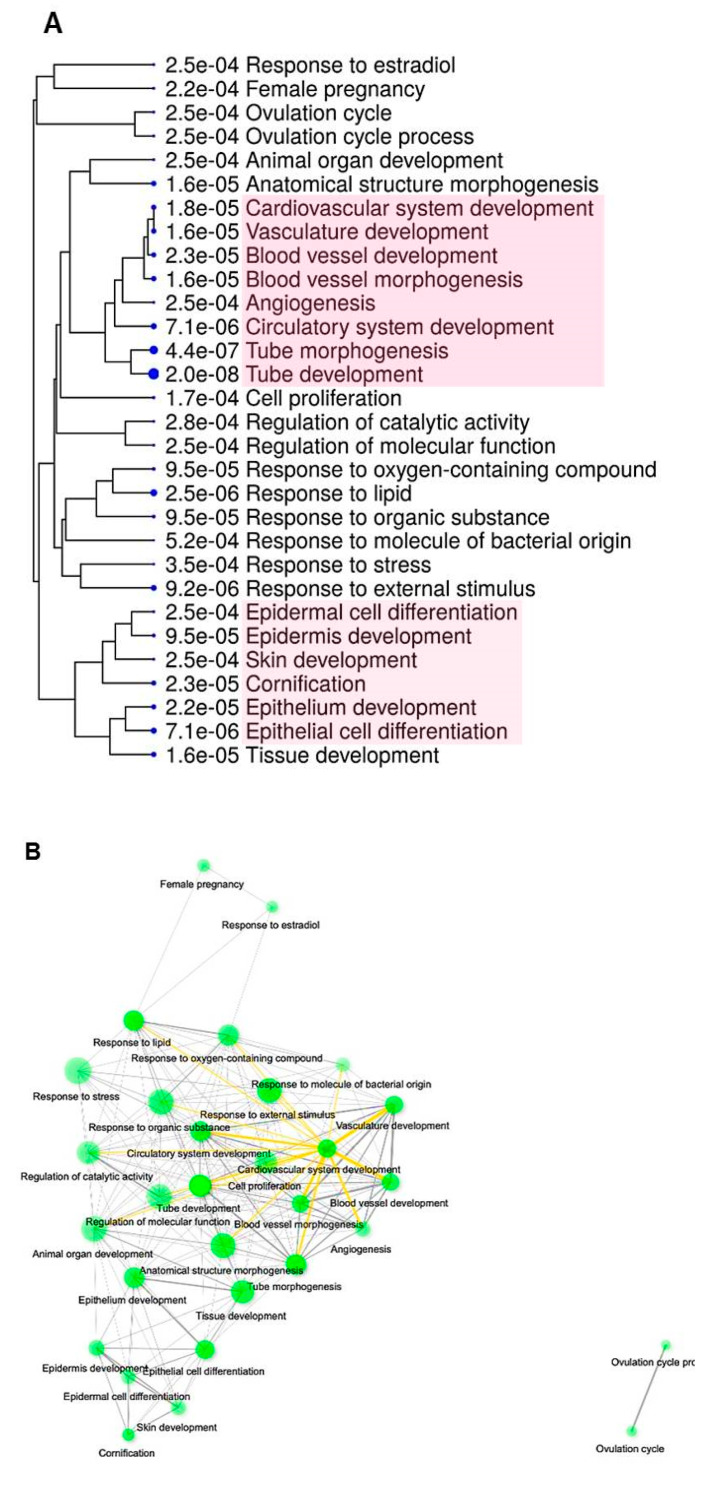
Enriched GO Biological Process (GO-BPs) terms for the 201 genes differentially expressed between early invasive CR and CIN. (**A**) Hierarchical tree summarizing significant GO terms (visualized using ShinyGO tool [52]). The size of blue dots at the end of branches corresponds to FDR values printed in front of the terms. Terms sharing more genes are grouped together. Vasculature and epithelium/epidermis development-related terms are grouped in two clusters and colored in pink; (**B**) The network of enriched GO-BPs terms. Brightness of the nodes represents the probability that a particular functional category is overrepresented in the dataset. Thickness of the lines indicates the percent of overlapping genes. Lines connecting the most prevalent pathway nodes are shown in yellow.

**Figure 12 ijms-21-06515-f012:**
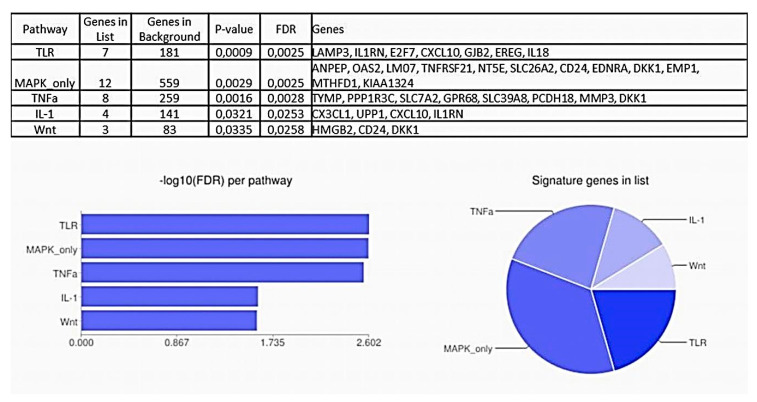
SPEED analysis of transcriptomic perturbations in early-stage invasive cervical cancer. Screenshot of the SPEED results is presented in the upper table; the bar chart describes statistical significance of pathway enrichment and the pie chart represents the distribution of signature genes in each pathway.

**Figure 13 ijms-21-06515-f013:**
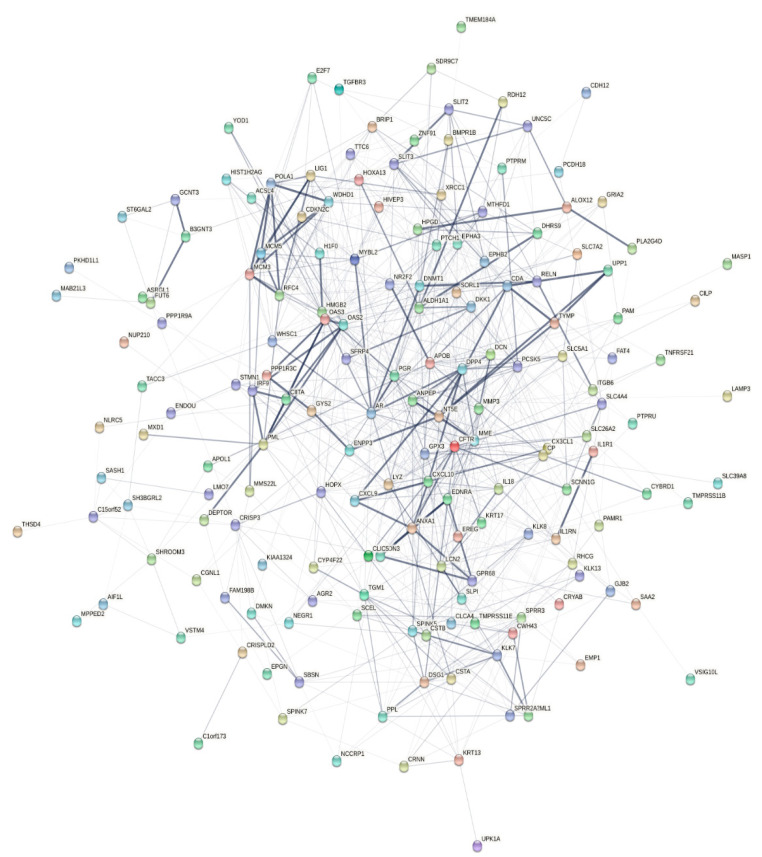
Visualization of the PPI network of DEGs by means of STRING and ShinyGO tools. Thickness of the lines represents statistical significance of predicted interactions between each pair of genes (the nodes shown as colored circles).

**Figure 14 ijms-21-06515-f014:**
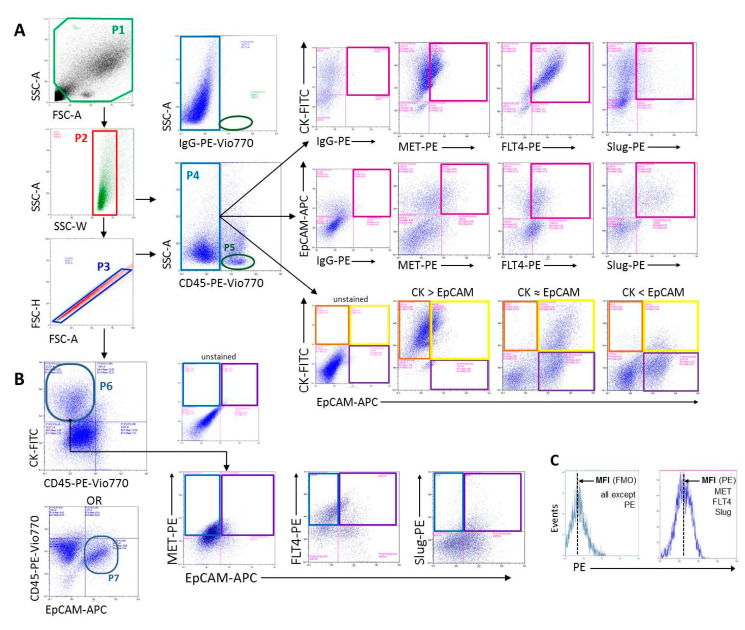
Gating strategy for identification of MET (HGFR), FLT4 (VEGFR3), and SLUG (SNAI2) expmune checkpoint mediators involved in regulati. (**A**) Gates P1–P3 were set up to exclude cell debris and cell aggregates, cell singlets identified in gates P2 and P3 (outlined with red and dark-blue rectangles) were plotted against CD45 and gate P4 was then drawn to identify CD45(-) non-immune cell population (blue rectangle). For P4-gated cells, cytokeratin (CK) or EpCAM were plotted against MET, FLT4, and SLUG, to assess the abundance of CD45 (-) CK (+) and CD45 (-) EpCAM (+) cells expressing these markers (purple rectangles) within the entire population of non-immune cells. CK was also plotted against EpCAM in order to define CK (+) EpCAM (+), CK (+) EpCAM (-), and CK (-) EpCAM (+) cell populations (yellow, orange and violet rectangles, respectively); three different patterns for co-expression of epithelial markers with the prevalence of one of these populations are shown. Gate P5 (outlined with green) drawn based on SSC/CD45 was used to enumerate %TILs (tumor-infiltrating lymphocytes); lymphocyte identity of cells measured in this gate was examined in a separate experiment based on common lymphocyte markers expression (particularly, CD3/4/8 T cell markers) using back-gating analysis (not shown); (**B**) CD45 (-) CK (+) cells from gate P6 (or CD45 (-) EpCAM (+) cells from gate P7) were further analyzed for co-expression pattern of another epithelial marker (EpCAM or CK) and lymphangiogenic/EMT markers MET, FLT4 and SLUG (violet and blue rectangles outline triple- and double-positive cell populations respectively). As the pan-cytokeratin-specific antibody used in this assay recognizes common epitopes in CK7, CK8, CK18, and CK19 robustly expressed in both simple and stratified epithelia, as well as in endothelial cells and some other of mesenchymal tissues [70,71,72,73,74,75], we assumed that CD45 (-) EpCAM (+) and CD45 (-) CK (+) populations isolated from cervical cancer tissues could be enriched in both tumor cells and vascular (blood/lymphatic) endothelial cells though may contain other cell types; (**C**) The relative increase in the Median Fluorescence Intensity (MFI) was calculated to evaluate the level of protein expression of the selected markers. IgG—isotype control antibodies. FMO—Fluorescence Minus One control.

**Figure 15 ijms-21-06515-f015:**
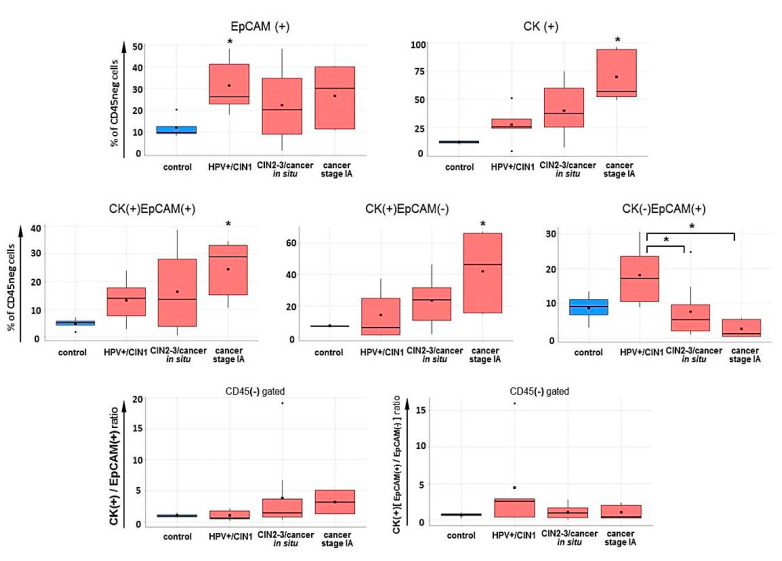
Boxplots showing flow cytometry results for pan-cytokeratin (CK) and EpCAM expression in cell suspensions derived from normal and neoplastic cervical epithelium. Asterisks (*) indicate significant differences between the patient sample group and the control group at *p* < 0.05 level (Wilcoxon Mann–Whitney *U*-test) or between two patient sample groups if indicated.

**Figure 16 ijms-21-06515-f016:**
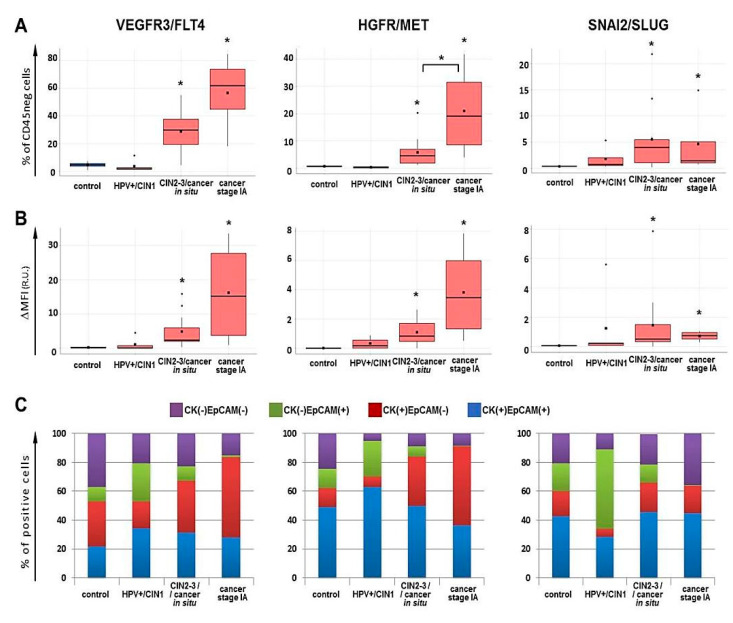
Flow cytometry results for FLT4/VEGFR3, MET/HGFR, and SLUG/SNAI2 expression in CD45 (-) cell population derived from dissociated cervical biopsies. (**A**) Boxplots showing the percentages FLT4/MET/SLUG expressing cells within CD45(-) population; (**B**) Boxplots showing the MFI relative increase calculated as Δ(MFI) = [MFI(FLT)-MFI(FMO)]/MFI(FMO), where FMO is Fluorescence Minus One control; (**C**) Histograms showing pattern of cytokeratin (CK) and EpCAM co-expression within CD45 (-) cells positive for FLT4, MET or SLUG expression. Asterisks (*) indicate significant differences between the patient sample group and the control group at *p* < 0.05 level (Wilcoxon Mann–Whitney *U*-test) or between two patient sample groups if indicated.

**Figure 17 ijms-21-06515-f017:**
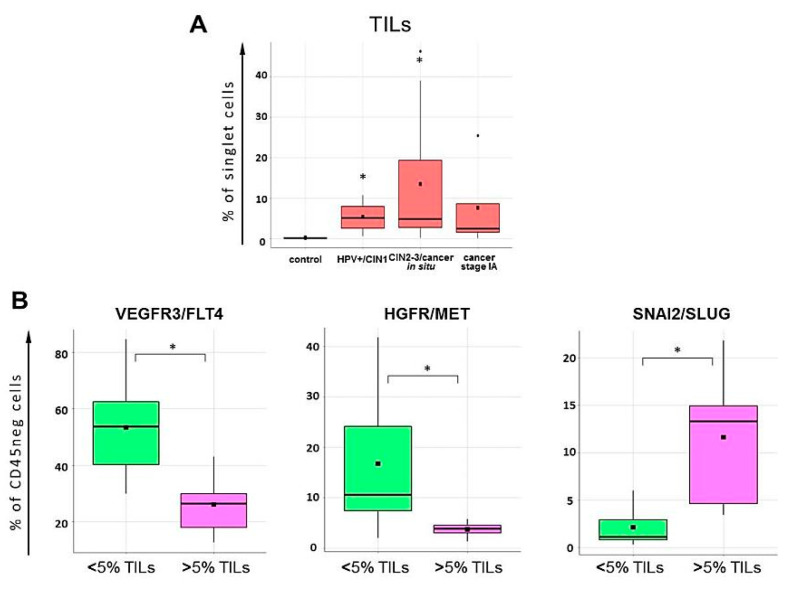
FLT4/VEGFR3, MET/HGFR, and SLUG/SNAI2 cell expression correlate with the level of immune (specifically, lymphocytic) infiltrate in cervical neoplastic locus. (**A**) Tumor-infiltrating lymphocyte (TILs) counts; (**B**) Abundance of FLT4/VEGFR3, MET/HGFR, and SLUG/SNAI2 expressing cells in CD45 (-) cell population derived from CIN3/cancer in situ and cervical cancer stage IA samples with relation to the degree of lymphocytic infiltrate; sample groups displaying the percentage of TILs <5% or >5% are shown as green and purple boxes, respectively. Asterisks (*) indicate significant differences between the groups at *p* < 0.05 level (Wilcoxon Mann–Whitney *U*-test).

**Table 1 ijms-21-06515-t001:** A panel of cervical tissue samples selected for RNA-Seq: designations and diagnosis [14].

Sample ID ^1^	Degree/Stage
Norm	morphologically normal cervical epithelium
CIN_1	cervical intraepithelial neoplasia grade 3 (CIN3)
CIN_2	CIN3 (carcinoma in situ, CIS)
CIN_3	CIN2/3 (high-grade squamous intraepithelial lesion)
CIN_4	CIN3 (CIS)
CR_1	invasive carcinoma at IA1 stage
CR_2	invasive carcinoma at IB1 stage
CR_3	invasive carcinoma at IA1 stage
CR_4	invasive carcinoma at IA1 stage
CR_5	invasive carcinoma at IB1 stage
CR_6	invasive carcinoma at IB2/IIA1
CR_7	invasive carcinoma at IIB stage

^1^ Each sample was obtained from a different patient.

**Table 2 ijms-21-06515-t002:** Enriched KEGG pathways.

Gene in List	Total Genes	Functional Category	Enrichment FDR	Genes
5	36	DNA replication	0.0011	*LIG1, POLA1, MCM3, MCM5, RFC2/4*
6	166	Influenza A	0.024	*CXCL10, OAS, IRF9, CIITA, IL18, PML*
7	181	Axon guidance	0.026	*EPHA, EPHB, SLIT2, SLIT3, UNC5, PTCH1, BMPR1B*
5	57	Pyrimidine metabolism	0.029	*ENPP3, NT5E, CDA, UPP1, TYMP*
4	69	Renin secretion	0.046	*EDN, EDNRA, CLCA, ADCYAP1R1*

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
