# Peer review of "Markers of Angiogenesis, Lymphangiogenesis, and Epithelial–Mesenchymal Transition (Plasticity) in CIN and Early Invasive Carcinoma of the Cervix: Exploring Putative Molecular Mechanisms Involved in Early Tumor Invasion"

_ijms, 2020, doi:10.3390/ijms21186515_

Round 1

Reviewer 1 Report

This manuscript by Kurmyshkina et al entitled “Markers of Angiogenesis, Lymphangiogenesis and Epithelial-Mesenchymal Transition (Plasticity) in CIN and Early Invasive Carcinoma of the Cervix: Exploring Putative Molecular Mechanisms Involved in Early Tumor Invasion” provides a comprehensive study of the putative genes and pathways that play a role in the transition from HPV+ CIN to early stage cervical cancer. This study provides a detailed look at cervical cancer biology at the stage between CIN and low grade cancer - providing data on the role of genes in EMP as well as inflammation. This study provides a basis for many interesting future studies in cervical cancer biology. Overall, the research performed in this paper was comprehensive and thorough with no major issues found in the research strategy or presentation of the data. I recommend minor grammatical changes as highlighted in the attached PDF. For example replacement of the word 'edges' with lines. Also, all of the symbols in the paper are missing such as 'delta', 'alpha' - TNF-alpha, 'beta', etc. In addition, the boxplot figures are difficult to read in the current color scheme – the brackets are not easily found nor some of the asterisks. I would recommend removing the blue background color on these plots which include figures 14, 15, 16 and in the supplementary figures.

Author Response

Response to Reviewer 1 Comments

Point 1: I recommend minor grammatical changes as highlighted in the attached PDF. For example replacement of the word 'edges' with lines. Also, all of the symbols in the paper are missing such as 'delta', 'alpha' - TNF-alpha, 'beta', etc.

Response 1:

First, we checked greek symbols across the text and inserted the missing ones on pages: p.17 (lines 402,422), p.18 (lines 434,447), p.23 (line 558), p.24 (line 570), p.28 (line 774), p.32 (lines 965,981), p.33 (line 1019) (it should be noted that these symbols were present in the original Word-version of the manuscript during submission; probably they were lost during subsequent format conversions).

 Then, the word ‘edges’ was replaced with ‘lines’ (on pages 17 and 20, lines 389 and 470) and the sentences on page 24 (lines 585-587) and p.27 (lines 727-730) were reformulated as suggested by Reviewer; the word ‘consensual’ in the legend to Figure 9 (old number - Fig.8) (p.14) marked by Reviewer was deleted as redundant and, indeed, unclear.

Point 2: In addition, the boxplot figures are difficult to read in the current color scheme – the brackets are not easily found nor some of the asterisks. I would recommend removing the blue background color on these plots which include figures 14, 15, 16 and in the supplementary figures.

Response 2: We redraw the boxplots in Figures 14, 15, 16 (new numbers - Fig.15,16,17), S1, and S2 using white background, colored boxes and larger size of markers as recommended.

Reviewer 2 Report

This is a very important paper stating on RNA-seq data and protein data (FACS, western) using cervical cancer patients samples, suggesting the significance of some biological signaling pathways in this pathological disease progression.

Some minor points to be addressed and considered;

  1. Among the signaling pathways described, PI3K signaling, TGF-b signaling and developing pathways are not listed (Fig. 11, page 17).  Some description might be necessary to state whether the pathways activation are not significantly detected or not.
  2. EPHA signaling (table 2, page 16 and page 17, page 27) is not only serving as cellular and axonal guidance molecule, but also functioning to transduce ECM (extracellular matrix) status (stiffness) inside cell (Fattet, et al; Develop Cell 2020), using its ligand-independent mechanism.  Maybe better to add this aspect.
  3. Schematic summary might be helpful for general audience to understand the whole information. Fig. 1 might be a general introduction and one additional schema summarizing this work would be necessary.

Author Response

Response to Reviewer 2 Comments

Point 1: Among the signaling pathways described, PI3K signaling, TGF-b signaling and developing pathways are not listed (Fig. 11, page 17).  Some description might be necessary to state whether the pathways activation are not significantly detected or not.

Response 1: We fully agree with the notion that PI3K signaling and TGFbeta-mediated signaling are fundamental cancer pathways widely known to play crucial role in cervical cancer progression. Concerning Fig.11, SPEED Database analysis of DEGs indeed didn’t find PI3K- and TGFbeta-dependent and developmental pathways to be statistically significant causal pathways leading to gene expression perturbations in our comparison of early-invasive versus pre-invasive cancer sample set (only pathways that fit adjusted cutoff level are listed as indicated in the text). We suppose this could be due to several reasons; one of them may be that activation of such multifunctional pathways as PI3K-dependent signaling occurs at earlier stage. At the same time, many members of PI3K-, TGFbeta-associated and developmental pathways were found among the DEGs (as mentioned in the above text, par.2.1.) so that they could be considered the significant downstream targets of pathways listed in Fig.11 (new number - Fig.12).

Point 2: EPHA signaling (table 2, page 16 and page 17, page 27) is not only serving as cellular and axonal guidance molecule, but also functioning to transduce ECM (extracellular matrix) status (stiffness) inside cell (Fattet, et al; Develop Cell 2020), using its ligand-independent mechanism.  Maybe better to add this aspect.

Response 2: We fully share the Reviewer’s opinion that non-canonical and ligand-independent activities (such as regulation of mechanotransduction pathways) and functions of ephrins, including EPHA, present an important aspect with many newly-emerging data worth of attention. We didn’t make changes to Table 2 (because it uses KEGG database terms), but we mentioned this aspect in the main text body (on pages 17 and 28, lines 399 and 751) citing the reference provided by Reviewer, since this new article also addresses the problem of carcinoma in situ-to-invasive cancer transition (numeration of references, beginning with [53], has been changed accordingly). Although this article reported specifically about EPHA2-mediated ECM status-sensing, we can assume this may be a more common feature of ephrins depending on biological process, tumor type, etc.

Point 3:    Schematic summary might be helpful for general audience to understand the whole information. Fig. 1 might be a general introduction and one additional schema summarizing this work would be necessary.

Response 3: We tried to provide an introductory explanatory Figure 1, summarizing the idea and the design of the present work (page 3). [Numeration of figures has been changes accordingly]. We hope that we properly understood the task, but if not, we’re ready to redo it (we assure that the figure or its elements were not adopted from any resource).
